# Global prevalence of elevated estimated pulmonary artery systolic pressure in clinically stable children and adults with sickle cell disease: A systematic review and meta-analysis

**Mobin Ghazaiean** [1,2]*, **Hadi Darvishi-Khezri**[3], **Behnam Najafi**[1], **Hossein Karami**[3], **Mehrnoush Kosaryan**[3]

**1** Student Research Committee, School of Medicine, Mazandaran University of Medical Sciences, Sari, Iran, **2** Gut and Liver Research Center, Non-communicable Disease Institute, Mazandaran University of Medical Sciences, Sari, Iran, **3** Thalassemia Research Center (TRC), Hemoglobinopathy Institute, Mazandaran University of Medical Sciences, Sari, Iran

\* dr.mobin.gh@gmail.com

## Abstract

### Background

The current study sought to determine the prevalence of elevated estimated pulmonary artery systolic pressure (ePASP) in clinically stable children and adults with sickle cell disease)SCD(worldwide.

### Methods

The studies included were identified through a search of databases such as PubMed, Scopus, Science Direct, Web of Science, and Embase, as well as Google Scholar engine, adhering to specific inclusion and exclusion criteria. Heterogeneity among the primary study results was assessed using the I-squared index, while publication bias was evaluated through funnel plots, Egger's test, and trim and fill analysis. All statistical analyses were conducted using R software, version 4.3.0.

### Results

79 primary studies were included, comprising 6,256 children (<18 years old) and 6,582 adults (≥18 years old) with SCD from 22 countries. The prevalence of elevated ePASP was found to be 21.8% (95% confidence interval [CI]: 18.46 to 25.07) in children and 30.6% (95% CI: 27.1 to 34.1) in adults. The prevalence of elevated ePASP among studies with severe SCD genotypes including HbSS and HbS/$\beta^0$ was found to be 19.45% (95% CI: 14.95 to 23.95) in children and 29.55% (95% CI: 24.21 to 34.89) in adults. Furthermore, sex-specific prevalence among SCD patients with elevated ePASP indicated the highest prevalence in male children at 60.35% (95% CI: 54.82 to 65.88) and adult female patients at 54.41% (95% CI: 47.3 to 61.5). A comparative analysis of the mean values of clinical

**Data availability statement:** "All relevant data are within the paper and its Supporting Information files."

**Funding:** The author(s) received no specific funding for this work.

**Competing interests:** The authors have declared that no competing interests exist.

**Abbreviations:** PASP, Pulmonary artery systolic pressure; ePASP, Estimated pulmonary artery systolic pressure; SPAP, Systolic Pulmonary artery pressure; SCD, sickle cell disease; CI, Confidence interval; HbS, Hemoglobin S; PH, Pulmonary hypertension; SCA, Sickle Cell anemia; TRV, Tricuspid regurgitant jet velocity; TTE, Trans-Thoracic echocardiography (TTE); NT-BNP, N-terminal prohormone of brain natriuretic peptide; 6MWD, 6-min walk distance; PAP, Pulmonary Artery pressure; RHC, Right-Heart catheterization; ASH, American Society of Hematology; ATS, American Thoracic Society; VOC, Vaso-Occlusive crises; ACS, Acute chest syndrome; BMI, Body mass index; WBC, white blood cell; Plt, Platelet; HbF, fetal hemoglobin; Hb, hemoglobin; LDH, Lactate dehydrogenase; MD, Mean difference; SMD, Standardized mean difference; HU, hydroxyurea; LV, Left ventricular; LAD, Left atrial diameter; LVED, Left ventricular end-diastolic diameter; CSSCD, Cooperative Study of Sickle Cell Disease; NHLBI, National Heart, Lung, and Blood Institute; AHA, American Heart Association; NO, Nitric oxide; HIF-α, Hypoxia-inducible factor; HCP-1, Heme carrier protein-1.

and laboratory results revealed significant differences in several characteristics, including age, oxygen saturation, hemoglobin levels, fetal hemoglobin, white blood cell counts, platelet counts, and reticulocyte counts between patients with elevated ePASP and those without, in both children and adult SCD populations.

## Conclusion

Our findings regarding clinically stable SCD patients highlight a high prevalence of elevated ePASP.

## Introduction

Sickle cell disease (SCD) is a type of hemoglobinopathy characterized by a single mutation in the β-globin chain, which results in the substitution of valine for glutamic acid at the sixth amino acid position. This alteration leads to the formation of abnormal hemoglobin known as hemoglobin S (HbS) [1]. The global prevalence of homozygous SCD is approximately 112 per 100,000 live births, with significant regional variations—ranging from 43.12 per 100,000 in Europe to 1,125 per 100,000 in Africa. Annually, it is estimated that around 300,000 children are born with this condition. Furthermore, the global mortality rate is reported to be 0.64 per 100 years of child observation, with Africa experiencing the highest rate at 7.3 per 100 years of observation [2]. As individuals with SCD transition into adulthood, they often experience a heightened prevalence of cardiovascular complications, such as pulmonary hypertension (PH), stroke, and diastolic as well as left ventricular dysfunction [3]. Pulmonary involvement in SCD is a significant contributor to morbidity and mortality, and it is increasingly recognized as a critical determinant of patient survival. This has become particularly evident in recent years, as advancements in childhood survival strategies have enabled a growing number of SCD patients to reach adulthood [4]. Emerging evidence consistently indicates that PH complicated by right heart failure is a significant contributor to mortality in adult patients, particularly those suffering from homozygous sickle cell anemia (SCA) [5,6].

PH often remains asymptomatic during its initial stages. However, even in more advanced cases, studies indicate that the elevation of pulmonary artery systolic pressures (PASP) in patients with SCD tends to be mild to moderate, particularly when compared to individuals with idiopathic or scleroderma-associated PH [7]. This highlights the significance of using non-invasive methods like trans-thoracic echocardiography (TTE) for early detection of PASP elevations, as indicated by a tricuspid regurgitant jet velocity (TRV) of ≥2.5 m/s [8,9]. Elevated TRV, no matter the reason, has consistently been linked to premature death in adults [5,10]. Nevertheless, no connection has been found with premature death in children [11,12]. The significant mortality rate linked to TRV elevation and PH in adults with SCD highlights the importance of early screening and intervention in children, as pathophysiological changes that can lead to clinically significant PH may be reversible in early life [13]. In a recent study on a large population, TRV was assessed along with the risk of death using unbiased receiver operator curve analysis. The study concluded that a TRV value equal to or greater than 2.5 showed the best sensitivity and specificity for identifying the group within the SCD population at highest risk of death [14]. TRV stands for estimation of right ventricle and pulmonary artery pressure (PAP), and a confirmed diagnosis of PH is determined through cardiac catheterization with a mean PAP exceeding 25 mmHg [15,16]. Repeated ECHOs showing increased peak TRV are essential before sending for right heart catheterization (RHC) with PH expert guidance as variations in TRV measurements can occur due to technical issues, anemia severity, or higher cardiac output [17].

In published guidelines for sickle cell, the American Society of Hematology (ASH) and the American Thoracic Society (ATS) have differing recommendations on screening echocardiography for PH. The existing ATS guidelines endorse the inclusion of echocardiogram screening. On the other hand, ASH recommendations advise not to conduct echocardiogram screenings without cardiopulmonary symptoms, abnormal cardiac examination, or history of pulmonary embolism [17,18]. Numerous studies have highlighted significant variability in the echocardiography screening programs for patients with SCD. However, since 2014 [19], there has been no formal meta-analysis assessing the prevalence of elevated estimated pulmonary artery systolic pressure (ePASP) in this population. In this systematic review and meta-analysis, we seek to consolidate existing knowledge on the prevalence of elevated ePASP in patients diagnosed with SCD. Our primary objective is to provide evidence that can inform tailored policies for the prevention of PH in both children and adult populations globally, while also supporting further research initiatives. Establishing an accurate prevalence rate across various subgroups is particularly vital, as it serves as a prognostic indicator for health system planning and policymaking related to morbidity and mortality in SCD.

## Methods

An ethical declaration is not necessary for this study, as it involves a systematic review and meta-analysis of previously published literature. The study procedures were carried out in alignment with the 2020 PRISMA (Preferred Reporting Items for Systematic Reviews and Meta-Analyses) guidelines [20], supporting information file (S1 file). This review was registered with PROSPERO, CRD42024560916.

### Systematic search

We carried out a review of existing literature to evaluate the prevalence of elevated ePASP in children and adult patients with SCD. Subsequently, we conducted a meta-analysis to determine the overall prevalence of elevated ePASP and to analyze its occurrence in different clinical situations. We conducted extensive electronic searches across various databases, including PubMed, Scopus, Web of Science, Embase, and Science Direct, as well as the Google Scholar engine.

### Search strategy

The keywords used for the search were carefully selected from the Medical Subject Headings (MeSH) database, literature reviews, and other relevant index terms. The search terms included 'Sickle cell disease', 'Sickle Cell Anemia', 'Sickle Cell Trait' 'Hemoglobin S Disease', 'pulmonary hypertension', 'pulmonary arterial pressure', 'PAP', 'pulmonary artery systolic pressure', 'PASP', 'Tricuspid regurgitant jet velocity', 'TRV', 'echocardiography', and all possible word combinations were tailored to the specific patterns of each database. Additionally, the search was enhanced by manually reviewing the reference lists of identified articles. Additional detail on the study search is provided in supporting information file (S2 file).

### Inclusion criteria

The study's inclusion criteria comprised of research conducted from January 1, 2000, to December 31, 2023, in the English language that examined elevated ePASP in both children and adult populations. This encompassed various observational studies, including cross-sectional, case-control, and cohort studies, as well as those reporting on the baseline prevalence of elevated ePASP. The criteria also included studies with accessible full texts that

confirmed elevated ePASP diagnoses in individuals with SCD as determined by echocardiography. Additionally, studies focusing on the measurement of elevated ePASP as either a primary or secondary objective were also considered.

## Exclusion criteria

The exclusion criteria for this study encompassed several factors: duplicate studies, research unrelated to the study's subject and objectives, studies with ambiguous methodologies, interventional studies that lacked baseline reports of elevated ePASP, experimental designs, case reports, and non-English studies. To mitigate the influence of elevated pulmonary pressures resulting from vaso-occlusive crises (VOC) or acute chest syndrome (ACS) [21], we excluded studies involving hospitalized patients, individuals presenting with cardiopulmonary symptoms, and retrospective chart reviews of hospital records. We focused our analysis on steady-state prospective screenings, excluding publications that did not specify clinically stable conditions during echocardiography. Studies reporting mean TRV values, rather than categorized data, were omitted due to the inability to derive prevalence estimates for elevated ePASP. In cases where multiple publications originated from the same study population, we selected the publication that provided the most comprehensive information. Prevalence estimates from open cohorts were derived from the most recent publication available. Furthermore, we excluded conference abstracts, protocols, books/book chapters, preprints, narrative or systematic reviews, as well as letters, news articles, opinions, and commentaries. A detailed list of studies excluded during the initial screening and full-text screening stages, along with brief justifications, is available in supporting information file (S3 file) and (S4 file), respectively.

## Selection process

All collected references were organized using reference management software, specifically EndNote X7 (Version 17), which facilitated the removal of duplicates after sorting the records. Initially, the eligibility of the studies was evaluated based on their titles and abstracts. Subsequently, two investigators, H.K. and M.K., independently assessed the full texts. Any discrepancies were addressed with the involvement of the third author, H.D.K.

## Data collection process

Two physicians, M.G. and B.N., conducted data collection individually using a standardized and structured data collection form. Any discrepancies in their findings were resolved through the participation of the third author, H.K. The following details were gathered from each study: Study populations [The population dichotomized into children or adults based on the mean age, with <18 years considered children], year of publication [The date of publication was dichotomized into the previous meta-analysis (studies published from 2000 to 2014) and current meta-analysis (studies published from 2015 to the end of 2023)], location [Country of origin was dichotomized into western studies (United States and Europe) or Non-Western studies (Africa, Middle East, South America and Caribbean)], study design, sample size (including studies with less than 100 and more than 100 patients), age at the time of study, sex, definition and cut-off of elevated ePASP, the number of elevated ePASP, age of patients with elevated ePASP, sex of patients with elevated ePASP, mild and moderate to severe definition/cut-off in children, mild definition/cut-off in adults, number of patients with mild and moderate to severe elevated ePASP in children, number of patients with mild elevated ePASP in adults, the most severe forms of SCD (HbSS and HbS/$\beta^0$), lack of blood transfusion in the preceding 3 months, and hydroxyurea (HU)-naïve studies. Furthermore, mean values of the clinical findings including age, body mass index (BMI), 6-min walk distance (6MWD),

and O2 saturation and laboratory findings such as hemoglobin (Hb), fetal hemoglobin (HbF), white blood cell (WBC) counts, platelet (Plt) counts, lactate dehydrogenase (LDH), and reticulocyte counts were compared in adults, stratified by elevated ePASP cut-off points or categories, between SCD patients with elevated ePASP to those without. Also, mean values of the clinical findings including age and O2 saturation and laboratory findings such as Hb, HbF, WBC counts, Plt counts, LDH, and reticulocyte counts were compared in children, stratified by elevated ePASP cut-off points or categories, between SCD patients with elevated ePASP and those without.

We determined elevated ePASP using either the TRV or ePASP cut-off values established in existing literature. The prevalence of elevated ePASP in each study was calculated by dividing the number of elevated ePASP cases by the total number of patients who underwent echocardiography. In most cases, our definition of elevated ePASP prevalence aligned with published findings; however, some studies reported this prevalence based on the number of echocardiograms that yielded measurable TRV. In the limited studies that reported TRV or ePASP values in categorical formats, we determined the prevalence of elevated ePASP by consolidating the higher value categories into a single classification. Because echocardiography can usually measure TRV in individuals with PH, we presented the prevalence of elevated ePASP among SCD patients based on the total number of patients with measurable TRV. Furthermore, for clinical and laboratory findings, we presented the results based on the availability of mean ± SD data for both elevated and non-elevated ePASP patients. In cases of missing data regarding the outcome of interest, we made an effort to address this issue during the risk of bias evaluation.

## Risk of bias assessment

In this research, the risk of bias assessment investigated independently by two researchers (M.G. and H.D.K), with any discrepancies resolved by the third author (M.K.). A modified version of the checklist prepared based on the Newcastle–Ottawa scale (NOS) checklist [22], the Joanna Briggs Institute (JBI) Critical Appraisal Checklist [23], our study design, and findings of the included studies. A higher number of "yes" answers indicate better study quality. Additional details on the risk of bias assessment can be found in supporting information file (S5 file).

## Definitions

Age: We categorized individuals as children or adults depending on the average age, with those under 18 years old being classified as children [19].

Severity of elevated ePASP: TRV values that were equal to or greater than 2.5m/s were labeled as having elevated TRV, while values below 2.5m/s were deemed as normal TRV. A velocity of 2.5m/s–2.9m/s, equating to a systolic pulmonary artery pressure (SPAP) of around 30–39 mmHg, indicated a mild increase in PASP, while values of 3.0m/s or higher, corresponding to a SPAP of about 40–45 mmHg, were seen as a moderate to severe increase [24,25]. Detailed information regarding the definition/cut-off values is given in supporting information file (S6 file).

HU-naïve studies: Refers to studies where patients did not receive treatment with HU.

Lack of blood transfusion: Refers to studies where participants who had not been given a blood transfusion in the 90 days leading up to their enrollment.

Severe genotype: As per the literature, SCD genotypes like HbSS and HbS/β⁰ thalassemia are regarded as the most severe SCD forms [26].

Location: Researches with origins in the United States and Europe are classified as Western countries, while those from Africa, the Middle East, South America, and the Caribbean are classified as Non-Western countries.

Date of publication: The publication date was split into two categories for analysis: studies published between 2000 and 2014, and studies published from 2015 to 2023. One particular study, which was included in the previous meta-analysis, was published in 2015 [11].

## Data analysis

All statistical processes operated using the statistical software R version 4.3.0. The packages of METAPROP were deployed to generate the pooled prevalence. The combined prevalence of elevated ePASP was calculated using the inverse variance method and random effect model (REM), which was reported alongside a 95% CI. The META package was also utilized to synthesize the overall values of standardized mean difference (SMD) and mean difference (MD) for quantitative test results such as clinical findings and laboratory tests. To earn mean and standard deviation, we used the methods outlined by Luo et al. and Wan et al. [27,28], where the included studies presented median and interquartile range or median and range. Statistical heterogeneity was assessed using the Q test $I^2$. In addition, publication bias was evaluated through funnel plot and Egger's test, accompanied by the trim-and-fill method to identify and correct potential publication bias. Subgroup analyses were also run based on the availability of sufficient data from eligible studies, including age of patients with elevated ePASP, gender, the severity of ePASP, genotype severity of SCD, lack of blood transfusion within the preceding three months, studies recruiting HU-naive patients, sample size with cut-off value of 100, geographic area, and publication date. Furthermore, a meta-regression approach was employed to identify the sources of heterogeneity. In this model, we included studies involving adult patients with SCA, those exhibiting mild elevated ePASP, female participants with elevated ePASP, and HU-naïve studies. For children studies, the model encompassed SCA patients, male participants with elevated ePASP, HU-naïve studies, those with moderate-to-severe elevated ePASP, and studies focusing on severe genotypes of SCD. As the criteria for defining elevated ePASP were largely consistent across the primary studies, the need to investigate this variable further for potential heterogeneity was eliminated. A sensitivity analysis was conducted to assess the robustness of the examined meta-data. A p-value less than 0.05 was set as a threshold for statistical significance.

## Result

### Study selection

After conducting searches in multiple databases using relevant keywords, we initially found a total of 8,107 primary studies. Duplicate articles were identified and removed using End-Note software, resulting in a decrease of 5,225 records. Afterwards, we carefully screened the titles and abstracts to remove irrelevant content, reducing the count by 2,710. Additionally, we excluded 93 studies for various reasons: 82 did not meet the inclusion criteria, 8 lacked sufficient data reporting, and 3 were review articles listed supporting information file (S4 file). Through a rigorous screening process and the application of defined inclusion and exclusion criteria, we ultimately included 79 primary studies in the systematic review and meta-analysis. Vital information was extracted from these articles, and the selection process is visually represented in the PRISMA diagram (Fig 1).

### Study characteristics

The studies reviewed in this article cover the period from 2000 to 2023 and consist of a total of 79 investigations carried out in both Western and Non-Western countries. The breakdown of these studies is as follows: 40 were conducted in western countries and 39 in non-western

**PRISMA 2020 flow diagram for updated systematic reviews which included searches of databases, registers and other sources**

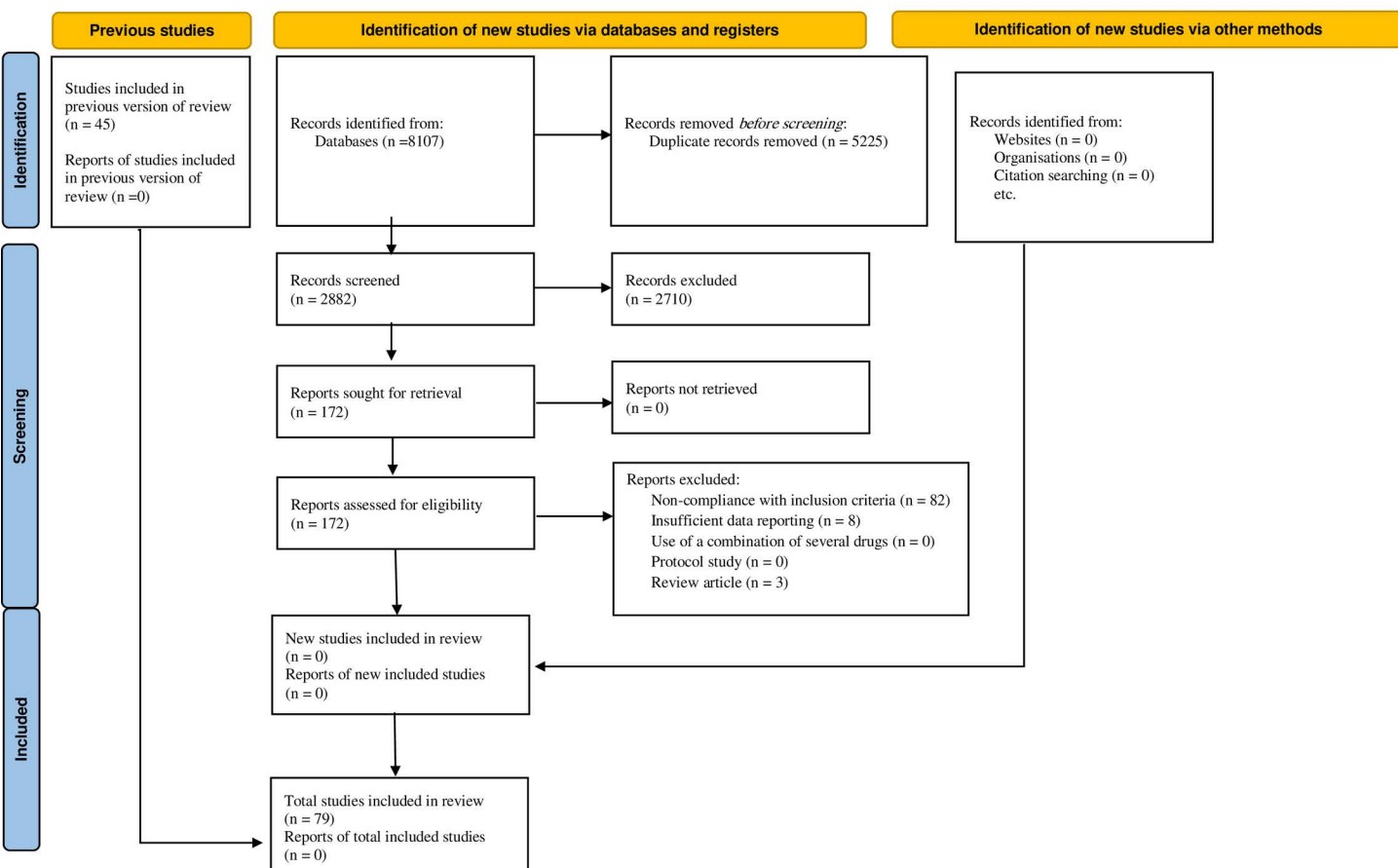

**Fig 1. Flow diagram of included/excluded studies.**

regions. Together, these studies involved 6,582 adult patients and 6,256 children diagnosed with SCD. Detailed characteristics of the included studies are provided in supporting information file (S6 file), eTables 1 and 2 for adults, and eTables 3 and 4 for children.

## Prevalence of elevated ePASP in children and adults

In 41 adult studies, the prevalence of elevated ePASP among individuals with SCD was examined. Of these studies, 21 were conducted in Western countries, while 20 originated from Non-Western regions. The reported prevalence of elevated ePASP varied significantly, ranging from 3.5% in the study by Ranque et al. to as high as 48.9% in the investigations by de Lima Marinho et al. and Olatunya et al. The heterogeneity indices revealed substantial variability among the primary studies, I-squared 96.1% (95% CI, 95.4 to 96.8), Q: 1038.84, P < 0.0001. Overall, the pooled results from these 41 studies indicate that the estimated prevalence of elevated ePASP among SCD patients is approximately 30.6% (95% CI, 27.1 to 34.1), as illustrated in Table 1 and Fig 2. Furthermore, the presence of publication bias is illustrated by the funnel plot diagram, supporting information file (S7 file, eFig 1) and corroborated by Egger's test, which yielded a bias estimate of 6.65 (SE = 0.70, P < 0.0001). To mitigate this bias, a trim and fill analysis was performed, resulting in the inclusion of 21 additional studies. Consequently, the overall prevalence of elevated ePASP among individuals is estimated to be 15.75% (95%

**Table 1. Summary of meta-analysis and subgroups analysis results of the elevated ePASP prevalence among adults with SCD.**

| Characteristics | No of studies/total | No of patients/total[c] | Prevalence rate | | Heterogeneity | |
|---|---|---|---|---|---|---|
| | | | ES (95% CI) | Model | Chi square | I square (%) |
| Age ≥ 18[a] | 13/41 | 561/1610 | 33.8 (30.4 to 37.2) | – | – | – |
| Sex[b] | | | | | | |
| Male | 11/41 | 148/327 | 45.6 (38.5 to 52.7) | Random | 17.82 | 43.9 |
| Female | 11/41 | 177/327 | 54.41 (47.3 to 61.5) | Random | 17.82 | 43.9 |
| Severity of elevated ePASP | | | | | | |
| Mild | 8/41 | 193/782 | 24.36 (21.36 to 27.36) | Random | 4.85 | 0 |
| Severe genotypes (HbSS and HbS/β0) | 16/41 | 650/2467 | 29.55 (24.21 to 34.89) | Random | 144.36 | 89.6 |
| Lack of blood transfusion in the preceding 3 months | 12/41 | 492/1869 | 31.24 (24.59 to 37.9) | Random | 89.96 | 87.8 |
| Hydroxyurea-naïve studies | 5/41 | 220/1030 | 25.61 (19.19 to 32.03) | Random | 17.4 | 77 |
| Sample size | | | | | | |
| <100 patients | 22/41 | 492/1419 | 33.6 (28.4 to 38.8) | Random | 183.51 | 88.6 |
| ≥100 patients | 19/41 | 1213/5163 | 27.6 (23.1 to 32.2) | Random | 733.43 | 97.5 |
| Location | | | | | | |
| Western countries | 21/41 | 1118/4019 | 31.24 (27.39 to 35.09) | Random | 236.86 | 91.6 |
| Non-Western countries | 20/41 | 587/2563 | 29.92 (23.81 to 36.04) | Random | 526.66 | 96.4 |
| Date of publication | | | | | | |
| Previous meta-analysis | 23/41 | 948/3138 | 31.24 (26.74 to 35.74) | Random | 296.39 | 92.6 |
| Current meta-analysis | 18/41 | 757/3444 | 29.87 (24.2 to 35.53) | Random | 538.35 | 96.8 |
| Overall | 41/41 | 1705/6582 | 30.6 (27.1 to 34.1) | Random | 1038.84 | 96.1 |

SCD: Sickle cell disease, SCA: Sickle cell anemia.

[a]The age of patients with elevated ePASP categorized into less than and more than 18 years old. Among the 41 studies, 13 studies reported the age of elevated ePASP patients, which was greater than 18 years old.

[b]The prevalence values based on the sex were calculated among SCD patients with elevated ePASP.

[c]This column represents the number of patients used to calculate the prevalence rate. Detailed information regarding the calculation of the prevalence values of each study is mentioned in S7 File. eFigures.

CI, 9.85 to 21.65). The heterogeneity indices were I-squared: 96.6% (95% CI, 96.1 to 97), Q statistic: 1774.04, P < 0.05. A meta-regression analysis was conducted to explore factors associated with heterogeneity. The results indicated that variables such as the presence of SCA studies (β = 0.07 [95% CI, −0.01 to 0.16], P = 0.08), mild elevated ePASP (β = 0.0002 [95% CI, −0.0031 to 0.0035], P = 0.92), female patients with elevated ePASP (β = −0.0022 [95% CI, −0.0063 to 0.0020], P = 0.30), and HU-naïve studies (β = −0.05 [95% CI, −0.16 to 0.06], P = 0.38) were not significantly associated with heterogeneity.

In 38 children studies, the prevalence of elevated ePASP among children with SCD was examined. Of these studies, 19 were conducted in Western countries and 19 in Non-Western countries. The reported prevalence of elevated ePASP in SCD patients varied significantly, ranging from 1.61% in the study by Odeyemi et al. to 55.6% in the research conducted by Zilberman et al. The heterogeneity indices (I-squared: 92.1% [95% CI, 90.1 to 93.7], Q: 470.3, P < 0.0001) indicated considerable variability among the primary studies. The pooled analysis estimates the overall prevalence of elevated ePASP among individuals with SCD to be 21.8% (95% CI, 18.5 to 25.1) as illustrated in Table 2 and Fig 3. Furthermore, publication bias is evident as indicated by the funnel plot diagram (eFig 2) and confirmed by Egger's test (Bias estimate: 3.92 [SE = 0.43], P < 0.0001). To mitigate this bias, a trim-and-fill analysis was performed, incorporating 19 additional studies. This adjustment led to an estimated overall prevalence of 10.47% (95% CI, 5.36 to 15.58). The heterogeneity indices were notably high, I-squared: 92.9% (95% CI, 91.5 to 94.1),

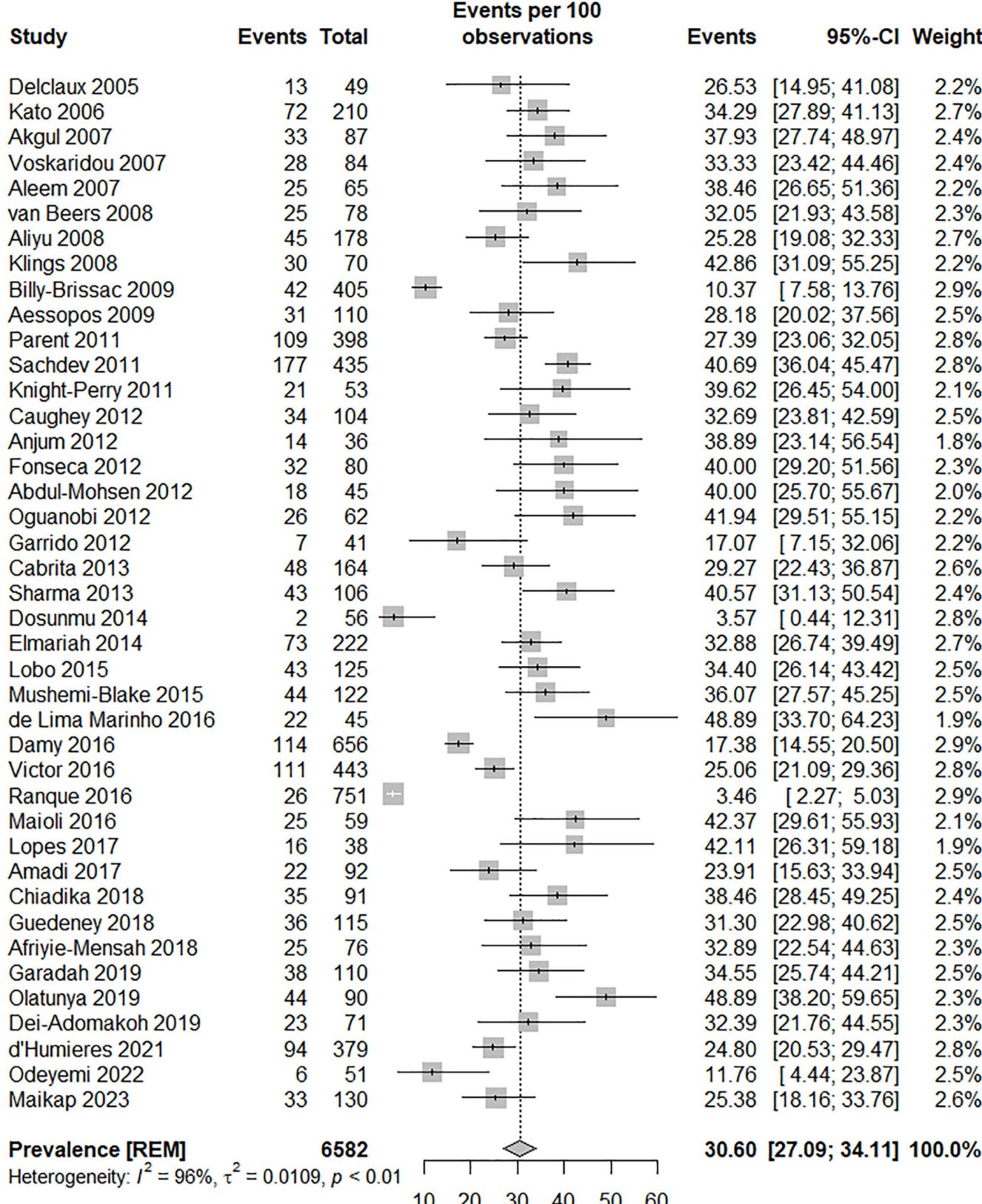

**Fig 2. The forest plot of the prevalence of elevated ePASP by the primary studies in adults, and the overall estimate (95% CI).**

**Table 2. Summary of meta-analysis and subgroups analysis results of the elevated ePASP prevalence among children with SCD.**

| Characteristics | No of studies/total | No of patients/total [c] | Prevalence rate | | Heterogeneity | |
|---|---|---|---|---|---|---|
| | | | ES (95% CI) | Model | Chi square | I square (%) |
| Age < 18[a] | 17/38 | 467/2238 | 22.6 (18.8 to 26.3) | – | – | – |
| Sex[b] | | | | | | |
| Male | 14/38 | 177/295 | 60.35 (54.82 to 65.88) | Random | 5.56 | 0 |
| Female | 14/38 | 118/295 | 39.65 (34.12 to 45.18) | Random | 5.56 | 0 |
| Severity of elevated ePASP | | | | | | |
| Mild | 7/38 | 98/577 | 18.7 (11.12 to 26.29) | Random | 40.53 | 85.2 |
| Moderate to severe | 6/38 | 28/537 | 5.82 (1.48 to 10.17) | Random | 24.95 | 80 |
| Severe genotypes (HbSS and HbS/β0) | 13/38 | 237/1276 | 19.45 (14.95 to 23.95) | Random | 54.91 | 78.1 |
| Lack of blood transfusion in the preceding 3 months | 6/38 | 95/571 | 17.93 (8.24 to 27.63) | Random | 66.31 | 92.5 |
| Hydroxyurea-naïve studies | 5/38 | 88/622 | 14.42 (9.02 to 19.83) | Random | 15.03 | 73.4 |
| Sample size | | | | | | |
| <100 | 28/38 | 399/1638 | 24 (19.9 to 28.1) | Random | 206.77 | 86.9 |
| ≥100 | 10/38 | 527/4618 | 16.8 (12.1 to 21.8) | Random | 191.44 | 95.3 |
| Location | | | | | | |
| Western countries | 19/38 | 512/2420 | 23.68 (19.03 to 28.32) | Random | 93.58 | 80.8 |
| Non-Western countries | 19/38 | 414/3836 | 19.84 (15.19 to 24.49) | Random | 196.25 | 90.8 |
| Date of publication | | | | | | |
| Previous meta-analysis | 22/38 | 545/2526 | 24.39 (19.95 to 28.83) | Random | 108.69 | 80.7 |
| Current meta-analysis | 16/38 | 381/3730 | 18.25 (13.59 to 22.91) | Random | 163.9 | 90.8 |
| Overall | 38/38 | 926/6256 | 21.8 (18.46 to 25.07) | Random | 470.3 | 92.1 |

SCD: Sickle cell disease, SCA: Sickle cell anemia.

[a]The age of patients with elevated ePASP categorized into less than and more than 18 years old. Among the 38 studies, 17 studies reported the age of elevated ePASP patients, which was less than 18 years old.

[b]The prevalence values based on the sex were calculated among SCD patients with elevated ePASP.

[c]This column represents the number of patients used to calculate the prevalence rate. Detailed information regarding the calculation of the prevalence values of each study is mentioned in S7 File. eFigures.

Q: 788.69, P < 0.0001. A meta-regression analysis was conducted to explore the factors contributing to heterogeneity. The results indicated that the presence of SCA studies (β= −0.06 [95% CI, −0.14 to 0.01], P = 0.1), male patients with elevated ePASP (β= 0.0034 [95% CI, −0.0041 to 0.0109], P = 0.37), HU-naïve studies (β= −0.821 [95% CI, −0.1731 to 0.0089], P = 0.08), moderate to severe elevated ePASP (β= 0.0105 [95% CI, −0.147 to 0.357], P = 0.41), and severe genotype of SCD (β= −0.0306 [95% CI, −0.1001 to 0.0389], P = 0.39) were not significantly related to heterogeneity.

## Prevalence of elevated ePASP based on the age and sex of the patients with elevated ePASP in children and adults

In 13 adult studies, age-specific prevalence was evaluated among SCD patients aged 18 years and older with elevated ePASP. The heterogeneity indices demonstrated substantial variability among these studies, I-squared: 51% (95% CI, 7.6 to 74), Q: 24.51, P-value: 0.02. When synthesizing the data from these 13 studies, the estimated prevalence of ePASP in this age group was found to be 33.8% (95% CI, 30.4 to 37.2) (Table 1).

In 17 children studies, age-specific prevalence was evaluated among SCD patients under the age of 18 with elevated ePASP. The analysis revealed significant heterogeneity among these studies, as indicated by the heterogeneity indices, I-squared: 76.2% (95% CI, 62 to

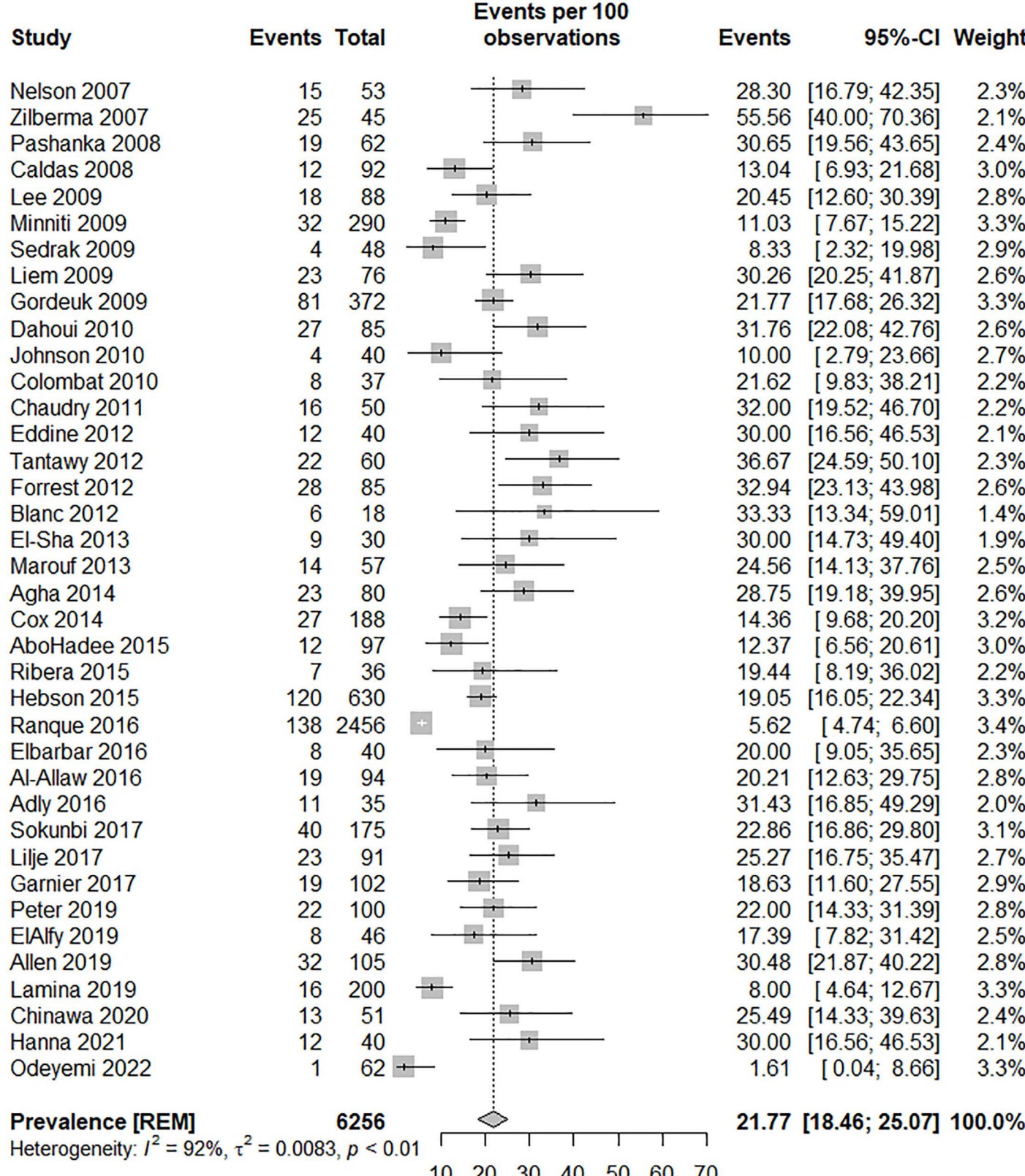

**Fig 3. The forest plot of the prevalence of elevated ePASP by the primary studies in children, and the overall estimate (95% CI).**

85.1), Q: 67.09, P < 0.0001). When synthesizing the results from these 17 studies, the estimated prevalence of ePASP in this age group is calculated to be 22.6% (95% CI, 18.8 to 26.3) (Table 2).

In 11 adult studies, sex-specific prevalence was evaluated among SCD patients with elevated ePASP. The heterogeneity indices revealed that numerically there was a trend towards some specific direction, but it was not statistically significant, I-squared: 43.9% [95% CI, 0 to 72.2], Q: 17.82, P = 0.06. When aggregating the results, the estimated prevalence of ePASP in male patients was found to be 45.6% (95% CI, 38.5 to 52.7) as illustrated in Table 1. Similarly, the analysis of female patients indicated an estimated prevalence of 54.41% (95% CI, 47.3 to 61.5), with the same heterogeneity indices reflecting that numerically there was a trend towards some specific direction, but it was not statistically significant, I-squared: 43.9% [95% CI, 0 to 72.2], Q: 17.82, P = 0.06, as shown in Table 1.

In 14 children studies, sex-specific prevalence was evaluated among SCD patients with elevated ePASP. The heterogeneity indices (I-squared: 0% [95% CI, 0 to 55], Q: 5.56, P = 0.96) revealed no significant heterogeneity among the results of these primary studies. Upon aggregating the data from these studies, the estimated prevalence of ePASP among male patients was found to be 60.35% (95% CI, 54.82 to 65.88), as detailed in Table 2. Similarly, the analysis of female patients with elevated ePASP also demonstrated non-significant heterogeneity (I-squared: 0% [95% CI, 0 to 55], Q: 5.56, P = 0.96). The combined findings indicated an estimated prevalence of 39.65% (95% CI, 34.12 to 45.18) among female patients, as illustrated in Table 2.

### Prevalence of elevated ePASP based on the severity of elevated ePASP in children and adults

In 8 adult studies, the prevalence of mild elevated ePASP has been assessed. The heterogeneity indices reveal non-significant variation among these studies, I-squared: 0% (95% CI, 0 to 67.6), Q: 4.85, P: 0.68. The aggregated data indicates that the estimated prevalence of mild elevated ePASP is 24.36% (95% CI: 21.36 to 27.36) (Table 1 and eFig 3).

In 7 children studies, the prevalence of mild elevated ePASP has been investigated. The heterogeneity indices reveal substantial variability among these studies, I-squared: 85.2% (95% CI, 71.4 to 92.3), Q: 40.53, P: < 0.0001. When synthesizing the results, the estimated prevalence of mild elevated ePASP is determined to be 18.7% (95% CI, 11.12 to 26.29), as illustrated in Table 2 and eFig 4. Similarly, six studies have provided insights into the prevalence of moderate to severe elevated ePASP. The heterogeneity indices for this subset also indicate significant variability, I-squared: 80% (95% CI, 56.5 to 90.8), Q: 24.95, P: 0.0001. The combined findings suggest that the estimated prevalence of moderate to severe elevated ePASP is 5.82% (95% CI, 1.48 to 10.17), as detailed in Table 2 and eFig 5.

### Prevalence of elevated ePASP based on the severe genotypes of SCD patients in children and adults

In 16 adult studies, the severe genotypes of SCD have been rigorously examined. The heterogeneity indices reveal substantial variability among these primary studies, I-squared: 89.6%, Q: 144.36, P < 0.01. Notably, the prevalence of elevated ePASP among individuals with severe SCD genotypes was found to be 29.55% (95% CI, 24.21 to 34.89) (Table 1 and eFig 6).

In 16 children studies, the heterogeneity indices indicated significant variability, with an I-squared: 78.1, Q: 54.91, P < 0.01. In this cohort, the prevalence of elevated ePASP among SCD patients with severe genotypes was determined to be 19.45% (95% CI, 14.95 to 23.95) (Table 2 and eFig 7).

### Prevalence of elevated ePASP based on the lack of blood transfusion in the preceding 3 months in children and adults

In 12 adult studies, the prevalence of elevated ePASP was assessed among individuals with SCD who had not received blood transfusions in the preceding three months. The heterogeneity indices revealed significant variability among the primary studies, I-squared: 87.8%, Q: 89.96, P < 0.01. The overall prevalence of elevated ePASP was found to be 31.24% (95% CI, 24.59 to 37.9) as detailed in Table 1 and eFig 8.

Additionally, a subset of 6 studies focused on the same population in children, with heterogeneity indices, I-squared: 92.5%, Q: 66.31, P < 0.01. The overall prevalence of elevated ePASP was found to be 17.93% (95% CI, 8.24 to 27.63), as presented in Table 2 and eFig 9.

### Prevalence of elevated ePASP based on the hydroxyurea-naive studies in children and adults

In five adult studies, the prevalence of elevated ePASP was evaluated in HU-naive adult patients with SCD. The heterogeneity indices revealed significant variation among these studies, I-squared: 77%, Q: 17.4, P < 0.01. The calculated prevalence of elevated ePASP stands at 25.61% (95% CI, 19.19 to 32.03) as detailed in Table 1 and eFig 10.

Similarly, in another set of five studies focusing on HU-naive children with SCD, the prevalence of elevated ePASP was also evaluated. The heterogeneity indices indicated notable variability, I-squared: 73.4%, Q: 15.03, P < 0.01. The prevalence of elevated ePASP in this group is reported at 14.42% (95% CI, 9.02 to 19.83), as shown in Table 2 and eFig 11.

### Prevalence of elevated ePASP based on the sample size in children and adults

In 41 adult studies, the prevalence of elevated ePASP was assessed, categorized by sample size. The heterogeneity indices for studies with fewer than 100 participants were recorded as I-squared: 88.6% and Q: 183.51, while those with more than 100 participants showed I-squared: 97.5% and Q: 733.43. The prevalence rates of elevated ePASP were found to be 33.6% (95% CI, 28.4 to 38.8) for studies with sample sizes under 100, compared to 27.6% (95% CI, 23.1 to 32.2) for those with sample sizes more than 100 (Table 1 and eFig 12). Notably, no significant difference was observed between these subgroups based on sample size (p-value: 0.09).

Similarly, in 38 children studies, the prevalence of elevated ePASP was also evaluated according to sample size. The heterogeneity indices for studies with fewer than 100 participants were I-squared: 86.9% and Q: 206.77, while for those with more than 100 participants, I-squared was 95.3% and Q: 191.44. The prevalence of elevated ePASP in studies with sample sizes less than 100 was 24% (95% CI, 19.9 to 28.1), whereas it was 16.8% (95% CI, 12.1 to 21.5) in studies with more than 100 participants (Table 2 and eFig 13). A statistically significant difference was observed between these subgroups based on sample size (p-value: 0.02).

### Prevalence of elevated ePASP based on the location in children and adults

In 41 adult studies, the prevalence of elevated ePASP has been carefully assessed across both Western and Non-Western countries. The heterogeneity indices for Western countries were reported as I-squared = 91.6% and Q = 236.86, while for non-Western countries they were I-squared = 96.4% and Q = 526.66. The prevalence of elevated ePASP was found to be 31.24% (95% CI: 27.39 to 35.09) in Western countries and 29.92% (95% CI: 23.81 to 36.04) in Non-Western countries (Table 1). Notably, there was no statistically significant difference between the subgroup results based on geographical location (p = 0.72).

In 38 children studies, the prevalence of elevated ePASP was investigated in children across both Western and non-Western countries. The heterogeneity indices for Western countries were recorded at I-squared: 80.8% and Q: 93.58, while non-Western countries exhibited I-squared: 90.8% and Q: 196.25. The prevalence rates of elevated ePASP were found to be 23.68% (95% CI, 19.03 to 28.32) for Western countries and 19.84% (95% CI, 15.19 to 24.49) for Non-Western countries (Table 2). Importantly, no statistically significant difference was noted between the subgroup results based on geographic location (p = 0.25).

### Prevalence of elevated ePASP based on the year of publication in children and adults

In 41 adult studies, the prevalence of elevated ePASP in adults has been systematically assessed according to publication dates. The heterogeneity indices for studies published in the previous meta-analysis were recorded as I-squared: 92.6% and Q: 296.39, while those for studies published after 2014 were I-squared: 96.8% and Q: 538.35. The prevalence rates of elevated ePASP were found to be 31.24% (95% CI, 26.74 to 35.74) for studies published in the previous meta-analysis and 29.87% (95% CI, 24.2 to 35.53) for those published thereafter (Table 1 and eFig 14). Importantly, no statistically significant difference was observed between the subgroups based on publication dates (p = 0.71).

In 38 adult studies, the prevalence of elevated ePASP in children has been assessed according to publication date. The heterogeneity indices for studies published in the previous meta-analysis were noted as I-squared: 80.7%, Q: 108.69, while those published after 2014 exhibited I-squared: 90.8%, Q: 163.9. The prevalence rates of elevated ePASP were found to be 24.39% (95% CI, 19.95 to 28.83) for studies published before 2015, and 18.25% (95% CI, 13.59 to 22.91) for those published thereafter (Table 2 and eFig 15), so numerically there was a trend towards some specific direction between the subgroup results based on the publication date, but it was not statistically significant (p = 0.06).

### Mean difference and standardized mean difference of the clinical and laboratory findings comparing elevated ePASP to those without among children and adults

Out of the original 79 studies, a significant number provided enough data for a comparative analysis of the mean differences in clinical and laboratory findings between SCD patients with elevated ePASP and those without. A random effects meta-analysis was conducted on the mean values of clinical parameters, including age, BMI, 6MWD, and O2 saturation, alongside laboratory parameters such as Hb, HbF, WBC counts, Plt counts, LDH, and reticulocyte counts in adults. This analysis was stratified according to various ePASP cut-off points or categories.

### Mean difference and standardized mean difference among adult studies

In adult patients with elevated ePASP, age [SMD = 0.62, 95% CI 0.27 to 0.97, P = 0.0005], LDH [SMD = 0.71, 95% CI 0.49 to 0.94, P < 0.0001], and reticulocyte counts [SMD = 0.3, 95% CI 0.07 to 0.53, P = 0.01] were significantly higher than patients without elevated ePASP. On the other hand, in adult patients with elevated ePASP, O2 saturation [SMD = −0.3, 95% CI −0.55 to −0.05, P = 0.02], Hb [SMD = −0.7, 95% CI −1.1 to −0.33, P = 0.0002], and HbF levels [SMD = −0.33, 95% CI −0.46 to −0.2, P < 0.0001] were significantly lower than patients without elevated ePASP. Detailed results of the MD and SMD for the clinical and laboratory findings can be found in Table 3 and supporting information file (S7 file, eFigs 16−33).

**Table 3. Summary of meta-analysis of the clinical and laboratory results among adults with SCD comparing elevated ePASP than those without.**

| Characteristics | No of studies/total | MD, 95% CI | Heterogeneity I square, 95% CI | P-value | P-value[k] | Model | SMD, 95% CI | Heterogeneity I square, 95% CI | P-value | P-value[l] |
|---|---|---|---|---|---|---|---|---|---|---|
| Age[a] (years) | 10/13 | 4.59, 3.43 to 5.74 | 17.6, 0 to 58.7 | 0.28 | **<0.0001** | Random | 0.62, 0.27 to 0.97 | 90.6, 84.9 to 94.2 | <0.0001 | **0.0005** |
| BMI[b] (Kg/m²) | 9/10 | 0.15, −0.69 to 0.98 | 77.6, 57.5 to 88.2 | <0.0001 | 0.73 | Random | 0.16, −0.21 to 0.54 | 89.6, 82.4 to 93.8 | <0.0001 | 0.39 |
| 6MWD[c] (meters) | 5/5 | −90.8, −194.8 to 13.1 | 98.5, 97.8 to 99 | <0.0001 | 0.09 | Random | −1.43, −3.05 to 0.19 | 93.3, 87.4 to 96.5 | <0.0001 | 0.08 |
| O2 saturation[d] (%) | 4/4 | −1.65, −2.55 to −0.74 | 0, 0 to 84.7 | 0.88 | **0.0004** | Random | −0.3, −0.55 to −0.05 | 0, 0 to 84.7 | **0.44** | **0.02** |
| Hb[e] (#/dl) | 12/12 | −0.92, −1.31 to −0.53 | 77.4, 60.7 to 87 | <0.0001 | **<0.0001** | Random | −0.7, −1.1 to −0.33 | 93.3, 90.1 to 95.5 | <0.0001 | **0.0002** |
| HbF[f] (%) | 8/10 | −1.27, −1.96 to −0.59 | 18.4, 0 to 61.2 | 0.28 | **0.0003** | Random | −0.33, −0.46 to −0.2 | 0, 0 to 67.6 | **0.62** | **<0.0001** |
| WBC[g] (×10³/ml) | 11/12 | 0.19, 0.03 to 0.35 | 2.7, 0 to 61.3 | 0.42 | **0.02** | Random | 0.07, −0.08 to 0.21 | 26.4, 0 to 63.6 | **0.19** | 0.39 |
| Plt[h] (×10³/ml) | 11/11 | −5.7, −11.8 to 0.46 | 0, 0 to 60.2 | 0.76 | 0.07 | Random | −0.07, −0.19 to 0.04 | 0, 0 to 60.2 | **0.59** | 0.22 |
| LDH[i] (U/l) | 10/12 | 197.1, 99.8 to 294.4 | 85.8, 75.7 to 91.7 | <0.0001 | **<0.0001** | Random | 0.71, 0.49 to 0.94 | 65.5, 32.4 to 82.4 | 0.002 | **<0.0001** |
| Retic[j] (%) | 9/9 | 1.2, 0.88 to 1.51 | 0, 0 to 64.8 | 0.96 | **<0.0001** | Random | 0.3, 0.07 to 0.53 | 76.3, 54.5 to 87.6 | <0.0001 | **0.01** |

SCD: Sickle cell disease, ePASP: estimated pulmonary artery systolic pressure, MD: mean difference, SMD: standardized mean difference, BMI: Body mass index, 6MWD: Six-minute walk distance, Hb: Hemoglobin, HbF: Fetal Hemoglobin, WBC: White Blood Cell, Plt: Platelet, LDH: Lactate dehydrogenase.

\*Total: Total number of studies reporting the clinical and laboratory results among adult SCD patients comparing elevated ePASP to those without.

\*\* No. of studies: The number of studies whose findings were able to be analyzed for calculating MD and SMD.

[a] Age: Median age of the dult patients with elevated ePASP is 38.5 years.

[b] Body mass index: Summary of the MD and SMD is provided in eFigs 16 and 17, respectively.

[c] 6-minute walk distance: Summary of the MD and SMD is provided in eFigs 18 and 19, respectively.

[d] Oxygen saturation: Summary of the MD and SMD is provided in eFigs 20 and 21, respectively.

[e] Hemoglobin: Summary of the MD and SMD is provided in eFigs 22 and 23, respectively.

[f] Fetal hemoglobin: Summary of the MD and SMD is provided in eFigs 24 and 25, respectively.

[g] White blood cell: Summary of the MD and SMD is provided in eFigs 26 and 27, respectively.

[h] Platelet: Summary of the MD and SMD is provided in eFigs 28 and 29, respectively.

[i] Lactate dehydrogenase: Summary of the MD and SMD is provided in eFigs 30 and 31, respectively.

[j] Reticulocyte count: Summary of the MD and SMD is provided in eFigs 32 and 33, respectively.

[k] P value corresponds to difference between MD of findings among subjects with elevated ePASP than those without.

[l] P value corresponds to difference between SMD of findings among subjects with elevated ePASP than those without.

**Table 4. Summary of meta-analysis of the clinical and laboratory results among children with SCD comparing elevated ePASP than those without.**

| Characteristics | No of studies/total | MD, 95% CI | Heterogeneity I square, 95% CI | P-value | P-value[i] | Model | SMD, 95% CI I square, 95% CI | Heterogeneity P-value | P-value[i] |
|---|---|---|---|---|---|---|---|---|---|
| Age[a] (years) | 16/17 | 0.41, −0.004 to 0.83 | 0, 0 to 52.3 | 0.57 | **0.05** | Random | 0.10, −0.01 to 0.22 | 0, 0 to 52.3 | **0.48** | 0.08 |
| O2 saturation[b] (%) | 8/9 | −1.17, −2.01 to −0.33 | 76, 51.9 to 88 | 0.0001 | **0.006** | Random | −0.71, −1.03 to −0.38 | 62, 17.9 to 82.4 | 0.01 | **<0.0001** |
| Hb[c] (g/dl) | 16/17 | −0.45, −0.66 to −0.24 | 54.4, 19.7 to 74.1 | 0.005 | **<0.0001** | Random | −0.37, −0.52 to −0.22 | 29.5, 0 to 61.4 | **0.13** | **<0.0001** |
| HbF[d] (%) | 7/9 | 1.1, 0.14 to 2.04 | 0, 0 to 70.8 | **0.46** | **0.02** | Random | 0.2, −0.04 to 0.45 | 42.8, 0 to 76 | **0.11** | 0.11 |
| WBC[e] (×10³/ml) | 10/11 | 0.67, −0.01 to 1.35 | 0, 0 to 62.4 | **0.77** | **0.05** | Random | 0.13, −0.02 to 0.29 | 0, 0 to 62.4 | **0.76** | 0.10 |
| Plt[f] (×10³/ml) | 8/8 | 60, 19.9 to 100.1 | 56.2, 3.5 to 80.1 | 0.03 | **0.003** | Random | 0.41, 0.12 to 0.7 | 60.7, 14.7 to 81.9 | 0.01 | **0.005** |
| LDH[g] (U/l) | 13/13 | 90.9, −11.4 to 193.2 | 94.7, 92.4 to 96.2 | <0.0001 | 0.08 | Random | 0.68, −0.27 to 1.62 | 86, 77.7 to 91.2 | <0.0001 | 0.16 |
| Retic[h] (%) | 14/14 | 3.1, 1.71 to 4.47 | 90.4, 85.7 to 93.6 | <0.0001 | **<0.0001** | Random | 0.86, 0.25 to 1.47 | 83.1, 73 to 89.4 | <0.0001 | **0.006** |

SCD: Sickle cell disease, ePASP: estimated pulmonary artery systolic pressure, MD: mean difference, SMD: standardized mean difference, Hb: Hemoglobin, HbF: Fetal Hemoglobin, WBC: White Blood Cell, Plt: Platelet, LDH: Lactate dehydrogenase.

\***Total:** Total number of studies reporting the clinical and laboratory results among children SCD patients comparing elevated ePASP to those without.

\*\* **No. of studies:** The number of studies whose findings were able to be analyzed for calculating MD and SMD.

[a]**Age:** Median age of the children patients with elevated ePASP is 13.5 years.

[b]**Oxygen saturation:** Summary of the MD and SMD is provided in eFigs 34 and 35, respectively.

[c]**Hemoglobin:** Summary of the MD and SMD is provided in eFigs 36 and 37, respectively.

[d]**Fetal hemoglobin:** Summary of the MD and SMD is provided in eFigs 38 and 39, respectively.

[e]**White blood cell:** Summary of the MD and SMD is provided in eFigs 40 and 41, respectively.

[f]**Platelet:** Summary of the MD and SMD is provided in eFigs 42 and 43, respectively.

[g]**Lactate dehydrogenase:** Summary of the MD and SMD is provided in eFigs 44 and 45, respectively.

[h]**Reticulocyte count:** Summary of the MD and SMD is provided in eFigs 46 and 47, respectively.

[i]P value corresponds to difference between MD of findings among subjects with elevated ePASP than those without.

[j]P value corresponds to difference between SMD of findings among subjects with elevated ePASP than those without.

## Mean difference and standardized mean difference among children studies

In children studies, a similar random effects meta-analysis was performed, focusing on the clinical findings such as age and O2 saturation, as well as laboratory findings including Hb, HbF, WBC counts, Plt counts, LDH, and reticulocyte counts. This analysis also compared SCD patients with elevated ePASP to those without, stratified by ePASP cut-off points or categories. In children with elevated ePASP, platelet counts [SMD = 0.41, 95% CI 0.12 to 0.7, P = 0.005] and reticulocyte counts [SMD = 0.86, 95% CI 0.25 to 1.47, P = 0.006] were significantly higher than patients without elevated ePASP. On the other hand, O2 saturation [SMD = −0.71, 95% CI −1.03 to −0.38, P < 0.0001] and Hb level [SMD = −0.37, 95% CI −0.52 to −0.22, P < 0.0001] in children with elevated ePASP were significantly lower than patients without elevated ePASP. Detailed results of the MD and SMD are presented in Table 4 and supporting information file (S7 file, eFigs 34–47).

## Risk of bias assessment

Out of 41 adult studies, 22 were categorized as having a medium risk of bias, while the number of studies classified as low and high risk of bias was 3 and 16, respectively. In the analysis of 38 children studies, 13 were identified as having a medium risk of bias, with 6 and 19 studies falling into the low and high-risk categories, respectively. Detailed assessments of the risk of bias can be found in supporting information file (S5 file).

## Discussion

This research systematically investigated the prevalence of elevated ePASP among clinically stable patients with SCD, as assessed through echocardiography in both children and adult populations. Consistent with expectations, we identified high prevalence of elevated ePASP, with no substantial differences between patients from Western and Non-Western countries. The high prevalence of elevated ePASP observed may be influenced by various clinical and laboratory factors. Prior research has demonstrated connections between the existence of PH and various clinical and laboratory findings [10,29–31], however a formal meta-analysis have not been conducted to accurately evaluate these findings in both children and adults with SCD at steady state. Clinical and laboratory analyses of children and adult patients with SCD revealed significant MD in various characteristics, including age, O2 saturation, Hb, HbF, WBC count, and reticulocyte count between those with elevated ePASP and those without. Considering the SMD values in adults, O2 saturation, HbF level, and reticulocyte count showed small to moderate effect, while age, Hb level, and LDH level demonstrated moderate to large effect. Moreover, the SMD values in children showed small to moderate effect for Hb level and Plt counts, moderate to large effect for O2 saturation, and large effect for reticulocyte count.

The relationship between TRV and age might illustrate the increased occurrence of heart disease that tends to rise with aging. As SCD patients grow older, the occurrence of elevated TRV and thus pulmonary vascular disease and right heart failure increases, heightening the mortality risk [32]. As age increases, left ventricular (LV) dilation and LV hypertrophy are anticipated to deteriorate, leading to progressive dilation of all 4 chambers. A recent meta-analysis on systolic function in SCD indicated that while patients with SCD exhibited normal LV systolic function, they experienced progressive LV dilation as time went on [33]. Even without SCD, it is recognized that PAP rises with age. In a chart review involving 3790 clinically normal echocardiograms showing measurable tricuspid regurgitation, ePASP was found to rise by approximately 1 mmHg for every decade of age [34].

Hemolysis associated with SCD, which releases free hemoglobin that scavenges nitric oxide, is thought to lead to endothelial dysfunction and intimal hyperplasia in the pulmonary

artery [35]. Hemolysis triggers platelet activation and hemostatic mechanisms in SCD and is believed to generate reactive oxygen species while stimulating vascular oxidases, processes that may cause both acute and chronic pulmonary vasoconstriction [36,37]. Clinical and translational research has identified links between hemolysis markers and endothelial dysfunction, elevated ePASP, as well as PH confirmed through RHC [5,38]. While not directly linked to hemolysis, increased WBC can be a significant source of oxidative stress in SCD [39,40], leading to NO dysregulation and changes in pulmonary vasculature [41,42]. Previous researchers have indicated a connection between the level of hemolysis, increased TRV, and reduced O2 saturation [5,32,43]. Like hemolysis, the level of anemia exhibited various independent clinical associations, such as increased measurements of left atrial diameter (LAD), left ventricular end-diastolic diameter (LVED), and TRV [44]. As hemoglobin levels drop, cardiac output has to rise to keep up with metabolic (oxygen) needs. With the rise in cardiac output, there is also an increase in the measured intracardiac velocities [11]. The cardiac characteristics of SCD patients, shaped by persistent anemia, hemolysis, and fluid overload, are marked by chamber dilation, eccentric hypertrophy, either normal or heightened left ventricular systolic function, and irregular diastolic filling [32].

Peripheral hemoglobin oxygen saturation, assessed noninvasively through pulse oximetry, is linked to various disease complications. Reduced O2 saturation has been linked to anemia, heightened reticulocyte levels [45], hemolysis [46], TRV [30] and diastolic dysfunction in SCD [47]. O2 saturation has an independent association LV hypertrophy and diastolic dysfunction in patients with SCD [47]. In research involving children with SCD, the LV E/E′ ratio was associated with oxygen saturation during both sleep and wakefulness. The key important correlations noted were that reduced sleeping and waking oxygen saturations relate to LV mass and diastolic dysfunction. These findings offer compelling support that low measurements of O2 saturation are linked to alterations in cardiac structure and function [47]. Slight reductions in steady-state oxygen saturation in SCD patients are related to the level of anemia, hemolysis indices, TRV, 6MWD, and the risk of stroke or other central nervous system occurrences [46–48]. Nonetheless, stable state desaturation is linked to greater desaturation during sleep [47,49], leading to the activation and adhesion of endothelial cells, leukocytes, and erythrocytes, events that may play a role in vasculopathy [50]. These results indicate that a thorough and methodical assessment of the clinical findings and laboratory results we currently obtain is crucial for enhancing care. It is crucial to distinguish the differences in findings between patients with and without elevated ePASP in each age group separately because it could indicate a specific group of patients requiring more attention in their regular care, especially in settings with limited resources. Regular echocardiograms can be more practical and economical in individual patient assessments when considering these clinical and laboratory findings for both children and adults age groups.

Our findings revealed that the prevalence of elevated ePASP among SCD patients is 30.6% in adults and 21.8% in children with SCD. In this study, 24.4% of adult patients exhibited mild TRV elevation. Four cohort studies evaluated the prevalence of PH through RHC among adult SCD patients with elevated ePASP, which the prevalence was 44%, 46%, 22%, and 23% in Fonseca et al. [51], Sharma et al. [52], Mushemi-Blake [53], and d'Humières [54], respectively. Among the children cohort, 18.7% showed mild TRV elevation, with 5.8% experiencing moderate to severe elevations. Although the increase in TRV observed in SCD patients is typically mild, it is nonetheless linked to structural remodeling of the right ventricle and right atrium, as well as elevated levels of NT-proBNP. Importantly, TRV elevations of ≥ 2.5 m/s have been associated with an increased risk of mortality [5,6,55]. Moderate to severe TRV elevation also correlates with a heightened risk of mortality and a greater likelihood of a definitive PH diagnosis through RHC [6,56,57]. The Pulmonary Hypertension and the Hypoxic Response in

SCD (PUSH) study indicated that a TRV of ≥ 2.7 m/second was observed in 4.9% of patients with SCD and was shown to have a significant correlation with mortality. The anticipated survival rate for the PUSH group of children, teenagers, and young adults with SCD was 99% up to 18 years, dropping to 94% by age 25 [58]. In a research involving 582 SCD patients in the US and UK, 22 fatalities were noted during the follow-up phase, with 50% exhibiting a TRV ≥ 3.0 m/sec. At 24 months, the total survival rate was 83% for TRV ≥ 3.0 m/sec and 98% for TRV < 3.0 m/sec [59]. Supported by a meta-analysis performed by Niss et al., drawing from various cross-sectional studies on both pediatric and adult groups, sickle cardiomyopathy initiates in childhood, with the first indication being left atrial dilation. The authors suggest that increased TRV stems from restrictive physiology and propose that individuals with sickle cardiomyopathy are at higher risk for lethal arrhythmias, akin to those with other types of restrictive cardiomyopathy. In conclusion, they determined that increases in TRV serve as indicators of this cardiomyopathy, posing a more significant issue in the medical treatment of SCD patients than the pulmonary hypertension itself [60]. Given that an increase in TRV is linked to anemia and cardiac remodeling seen in TTE, these situations elevate the risk for patients regarding anemia-induced organ harm and premature death [61]. Elevations in TRV indicate contributions from unusual pulmonary vascular resistance and LV diastolic impairment [62,63]. Diastolic dysfunction of the left ventricle, frequently seen in patients with SCD, may also lead to medial hypertrophy of the pulmonary artery by causing pulmonary congestion and increased venous pressure downstream. The existence of increased right ventricular pressure and left ventricular diastolic dysfunction are both independent and additive risk factors for mortality in older adults [9].

It is still unclear which screening approaches are effective for detecting PH in children with SCD. The 2014 ATS clinical practice guidelines suggest that ECHO screening should be done in all children as a way to establish baseline data, with potential NT-proBNP measurement for further support [18]. The guidelines from the National Heart, Lung, and Blood Institute (NHLBI) in the same year did not have enough data to make recommendations, but emphasized the importance of addressing symptoms or an abnormal ECHO [64]. The 2015 guidelines from the American Heart Association (AHA) and ATS suggest that pediatrics should have a regular ECHO screening by the age of 8, or sooner if showing symptoms, without specifying what those symptoms might be. It is also advised to consider NT-proBNP and BNP measurements [15]. On the other hand, the ASH guidelines for 2019 do not suggest regular ECHO screenings for asymptomatic children, except for those displaying particular symptoms regardless of age, and for those with suspicious ECHO results based on NT-proBNP levels [17]. ATS has pointed out several important clinical questions about PH in sickle cell lung disease that still need to be addressed [65]. Even with improvements in treatment, death rates remain alarmingly high among children, teenagers, and young adults with SCD in the United States. In a recent study in the US, the mortality rate up to age 18 was 6.1% for individuals with SCA and 1.6% for those with less severe forms of SCD [66]. Significantly, 17 out of 38 research studies involving children indicated the age of patients with elevated ePASP, with a median age of 13.5 years. Therefore, identifying the optimal age for screening is difficult, and clinical suspicion should remain high even in younger patients.

Most of the studies focused on patients with severe genotypes of SCD and the remaining studies comprised a mix of severe and less severe genotypes, with a notable predominance of severe genotypes. This could suggest that the overall prevalence in clinically stable SCD patients is somewhat overestimated. This finding is consistent with existing knowledge, as SCA patients typically experience more pronounced chronic hemolysis [67] which release of free plasma hemoglobin, could play a role in the PH onset owing to pulmonary vasoconstriction and vasoproliferation [68,69]. Recent studies suggest that endothelial nitric oxide (NO)

is diminished due to the release of toxic byproducts such as free plasma hemoglobin and arginase-1 from lysed red blood cells. The hemoglobin-induced scavenging of NO leads to the transcriptional upregulation of adhesion molecules and the expression of endothelin-1, a potent vasoconstrictor. Notably, increased concentrations of endothelin-1 have been detected in the plasma of patients with SCD [70]. Furthermore, research indicated that chronic anemia in SCA leads to sustained up-regulation of the hypoxic response, primarily through the activation of hypoxia-inducible factor (HIF-α), which subsequently results in damage to the pulmonary vascular endothelium [71]. In addition, splenectomy, thromboembolism, and sleep-disordered breathing have been linked to elevated ePASP in patients with SCD [72].

Among SCD patients with elevated ePASP, the prevalence of elevated ePASP was higher among males than females in children age range. There are contradicting reports in literature regarding whether PAH is more prevalent in males or females. Males have a higher susceptibility to hemolysis, and exhibit amplified pulmonary pressure and remodeling. The lungs of males show an increase in the uptake of free heme due to the upregulation of heme carrier protein-1 (HCP-1). Therefore, the activation of heme influx leads to intracellular heme signaling. Furthermore, mechanisms causing endothelial barrier leakage have also been documented [73,74]. These mechanisms were discovered to be increased in males specifically in the initial phase of the illness. Males' tendency towards hemolysis and activation of heme pathways may be due to higher levels of reactive oxygen species (ROS) production, causing increased fragility of red blood cells, while females are protected by the antioxidant properties of estrogen [75,76]. However, the prevalence of elevated ePASP was higher among adult females which may be attributed to factors like anemia [77], comorbid conditions like obesity [78], and the contraceptive pill and hormone replacement therapy [79,80]. Combining these findings reveals that estrogen alone is not responsible for the varying PAH phenotype observed in males and females. Lately, there has been a growing body of evidence suggesting that genes found on both the X and Y chromosomes play a role in the sexual differences seen in PH [81].

There were no significant differences in the prevalence of elevated ePASP between Western and Non-Western countries in both children and adults. The peak incidence of PH differs from one area to another. The differences in prevalence may also indicate the varied levels of organ damage and survival shown through various phenotypic expressions [82]. Estimating the prevalence of PH in developing countries is challenging due to socio-cultural, ethnic, geographic, and economic diversity, as well as regional variations in infrastructure for human development and healthcare. In situations where access to proper healthcare services is limited in developing countries, patients and doctors often fail to correctly diagnose illnesses. Although it has restrictions, echocardiography is not easily accessible in numerous facilities in developing nations [83]. People with SCD in developed nations can access a wide array of care services like HU and blood transfusions. In low-income countries, these interventions are not yet the usual care [84,85]. Due to the limitations of the included studies, we could only estimate the prevalence of elevated ePASP in HU-naïve patients, as the patients in other studies were heterogeneous in terms of receiving HU. Additionally, when assessing patients receiving HU, it is important to take into account potential confounding variables such as the proper dosing of HU, the length of HU therapy, and adherence to its use. It is worth mentioning that only two studies each in children and adults in Western countries after 2015 have reported clinical and laboratory findings of patients with elevated ePASP, making it difficult to directly compare Western and Non-Western countries.

The prevalence of elevated ePASP in children decreased slightly after 2015 compared to the meta-analysis conducted before 2015 [19]. The severity of SCD complications differ depending on variations in hemoglobin haplotype, HbF, α-thalassemia, Glucose-6-phosphate dehydrogenase deficiency, and UGT1A1 promoter polymorphisms [26]. The α-thalassemias are

commonly found in areas affected by malaria in Southeast Asia, Africa, India, and the Middle East. The α-globin genes are present in duplicate on each chromosome 16, resulting in a total of four α-globin genes (αα/αα). Deletions (e.g., $-\alpha^{3.7}$ and $-\alpha^{4.2}$) or mutations in genes (e.g. $\alpha^{\text{Constant Spring}}$ and $\alpha^{\text{TSaudi}}$) lead to a reduced synthesis of α-globin chains [86,87]. A significant occurrence of the 3.7 kb a-globin gene deletion has been noted in among SCA patients in Brazil (29%) [88], in India (32%) [89], in the UK among African Britons (34%) [90], in Guadeloupe (36%) [91], in Saudi Arabia (40%) [92], in the USA among African Americans (41%) [93], in France among Africans (48%) [94], and in Tanzania (58%) [95]. Taking into account the disparity in prevalence rates between the previous and current meta-analysis, it is important to note that almost 80% of children studies published after 2015 originated from Non-Western regions, while merely 25% of studies published prior to 2015 came from Non-Western countries. α-Thalassemia seems to diminish the disease's clinical severity by decreasing the mean corpuscular hemoglobin concentration, the proportion of dense cells, the degree of hemolysis, and the number of irreversibly sickled cells, while increasing total hemoglobin and hemoglobin A2 levels [96,97]. In the PUSH cohort of children, patients with α-thalassemia exhibited higher hemoglobin levels and lower LDH levels compared to those without, which aligns with α-thalassemia's role in preventing severe hemolysis and significant anemia. Moreover, the LV end diastolic Z score and LV mass index were higher in SCD patients without α-thalassemia [58]. Regarding cardiopulmonary complications, α-thalassemia was linked to a reduced prevalence of acute chest syndrome history, oxygen desaturation, increased TRV, and higher NT-proBNP levels in mostly adults from the Walk-PHaSST cohort; however, these patterns were less distinct in children from the PUSH cohort [58,59].

HbF significantly impacts the severity of SCD, particularly in those with HbSS and HbSβ0 genotypes [98]. Earlier clinical observations indicate that elevated levels of HbF have advantageous effects [99,100], such as increased hemoglobin concentrations and reduced hemolysis [101,102]. The level of protection provided by HbF depends on both the proportion of total Hb that is HbF and how HbF is distributed among all red blood cells [103]. The differences between individuals in HbF account for a substantial portion of the phenotypic variability observed in SCD. This difference in HbF is mainly influenced by genetics [104]. In an earlier investigation, organ damage appeared to be reduced in individuals with HbF levels surpassing 10%, while painful episodes and lung complications were lessened at levels exceeding 20% [100]. In the Cooperative Study of Sickle Cell Disease (CSSCD), levels of HbF were found to have an inverse relationship with the likelihood of painful crises, acute chest syndrome, and early death [105]. Several genetic and non-genetic factors can affect HbF levels in SCD patients, such as age, sex, the existence of alpha-thalassemia, genetic elements associated with the beta-globin gene locus, and additional genetic influences [96]. The complex interactions between variables influencing HbF levels in those with SCD could affect HbF's usefulness as a marker for cardiopulmonary problems. In general, the relationship between α-thalassemia and HbF concerning cardiopulmonary complications remains poorly defined [106]. Among children studies published after 2015, only two studies mentioned the HbF level among SCD patients with elevated ePASP, making it impossible to compare HbF level between earlier meta-analysis studies and those published after 2015. Numerous newly identified genetic modulators include polymorphisms in Endothelin-1 [107], and various other genes related to vascular function and NO signaling [108], which could be connected to the onset of PH. This is a field of growing research.

Several factors must be considered while interpreting the prevalence of elevated ePASP after 2015. While a 2D echocardiogram can be a useful preliminary test, it might fall short in diagnosing certain conditions due to its constraints, including subtle early disease signs and the chance of inexperienced operators missing a diagnosis during routine assessments,

especially considering that 80% of children studies conducted after 2015 were from Non-Western countries. Therefore, PH might be underdiagnosed, which poses a concern [109]. Furthermore, the increased occurrence of SCD in Non-Western countries could explain the differing outcomes observed between the previous meta-analysis and the current one [110], as approximately 75% of children studies carried out before 2015 and 80% of those performed after 2015 were from Western and Non-Western regions, respectively. Moreover, based on the earlier meta-analysis, a study involving 134 SCD patients in steady state, primarily children and young adults with a median age of 11 years (ranging from 3 to 22 years), where 76% had HbSS, revealed that a considerable portion exhibited left atrial enlargement (62%) and additional echocardiographic indicators of diastolic dysfunction. Every child exhibited normal systolic function. In contrast to classic restrictive cardiomyopathy, SCD typically features nearly universal LV enlargement due to the hyperdynamic circulation associated with chronic anemia, presenting a distinctive type of cardiomyopathy characterized by both restrictive and hyperdynamic physiology [60]. Among children studies published after 2015, only two studies mentioned the echo complications among SCD patients with elevated ePASP [111,112], making it unfeasible to compare echo complications between earlier meta-analysis studies and those published after 2015. Last but not least, A few children studies published after 2015 mentioned the clinical findings and laboratory data on patients with elevated ePASP, making it difficult to make a direct comparison between the two publication dates. Therefore, incorporating all of this information is difficult, and we are definitely still lacking important pieces of information.

Performing echocardiography to measure TRV can serve as a screening method for PH by estimating PAP [26]. For the analysis, elevated ePASP was determined based on the TRV or ePASP thresholds found in existing studies [19]. The majority of the studies included in this analysis utilized a TRV cut-off of 2.5 m/s to define elevated ePASP. Multiple factors can influence the TRV in patients with SCA. Echocardiography may not always detect PH due to the inability to measure TRV in some cases [113]. Acute but transient elevations in TRV have been noted during uncomplicated pain episodes or ACS potentially indicating temporary systemic changes [21]. This scenario is unlikely for our enrollees, all of whom were in a steady state condition at the time of enrollment. Additionally, TRV may overestimate PAP in SCA patients due to factors such as chronic anemia, compensatory increases in cardiac output, and volume overload, which can affect echocardiographic measurements [53,114]. The PASP is determined from the TRV using the modified Bernoulli equation $\left(PASP = 4\left(TRV\right)^2 + Right\ Atrial\ Pressure\right)$. Minor differences in TRV measurement can lead to notable variations in estimated pulmonary pressures [113]. Taking into account the errors in estimating RAP and the magnification of measurement errors from derived variables, the guidelines suggest using peak TRV (instead of estimated sPAP) as the primary variable for determining the echocardiographic likelihood of PH [115,116]. The effectiveness of using a peak TRV of $\geq 2.5$ m/s on screening ECHO alone to identify PH or decide on RHC may not be ideal, but can be enhanced by including other indicators such as reduced 6MWD or increased NT-BNP, or symptoms indicating PH [17]. A high-quality echocardiographic assessment by PH specialists is essential to decrease variability among observers [117].

Our study was designed to minimize false-positive diagnoses of elevated ePASP, particularly for cases deemed moderate to severe risk. The high prevalence of elevated ePASP identified in this review, particularly within this specific population, highlights the imperative for clinicians, policymakers, researchers, and stakeholders to focus more intently on elevated ePASP. Early diagnosis and timely access to care are crucial for enhancing the clinical trajectory of diseases and potentially alleviating their overall burden, particularly within healthcare systems that are under-resourced. The concerning outcomes observed in children with SCD

underscore the importance of early screening and the potential for preventive therapies to avert premature mortality in adults with SCD. Our research presents significant implications for future investigations into the etiologies and treatment of PH in patients with SCD.

## Limitations and strengths of the review

This review presents several limitations that warrant attention. First of all, despite our extensive search across multiple databases, the findings are derived from a relatively small number of original studies conducted within specific populations. Based on the included studies, we limited and focused our analysis to SCD patients at steady state during echocardiography, so we are unable to clarify how many patients were symptomatic. Moreover, while we endeavored to analyze distinct study populations, it was not always possible to ascertain independence among them. Another point that needs to be considered is the limited number of studies included in this review constrains our ability to conduct meaningful subgroup analyses to explore sources of heterogeneity, such as BMI, splenectomy, and less severe genotypes of SCD (HbSC and HbS/β+thalassaemia). Furthermore, as this is a meta-analysis of non-randomized studies, it is subject to the inherent limitations of observational data, including potential selection bias and unmeasured confounding variables. Importantly, the lack of representation from all countries within the specific populations may limit the generalizability of the results. Diastolic dysfunction of the left ventricle was proven to be an independent factor for the development of PH in SCD [9]. Some studies discussed complications seen on echocardiograms, such as diastolic dysfunction, while others did not address these issues, and the majority did not provide a clear definition of echo complications in individuals with elevated ePASP. Therefore, the studies included do not allow for a definitive assessment of echo complications in patients with elevated PASP. In addition, while some studies reported mortality rates during the study period, many did not, potentially leading to an overestimation of the prevalence rate. Another point that needs to be considered is that the TRV cutoff of ≥2.5 m/s seems inappropriate. Relying solely on TRV as a single surrogate marker may lead to a false negative rate of 33–42% in diagnosing PH in SCD patients, caused by inadequate Doppler TRV tracings and variability in both intraindividual and intraobserver TRV [118,119]. In patients with TRV of ≥2.5 m/s, including NT-BNP and 6MWD may enhance the accuracy of PH diagnosis (It is important to note that NT-pro-BNP levels can be deceptive in a condition of renal insufficiency) [18]. What's more, the reliability of a single echocardiographic measurement of TRV in the studied population remains unvalidated. Most studies failed to incorporate the mean of three TRV measurements when assessing elevated ePASP, resulting in variability in outcome precision. Furthermore, echocardiography is inherently subjective, relying on the examiner's skill, and many assessors were not blinded in both children and adult studies. Another limitation that needs to be addressed, the included studies exhibited heterogeneity in the administration of HU, lacking data on elevated ePASP in patients treated with HU versus those who were HU-naive. Besides, we were unable to confirm the increased prevalence of ePASP through RHC. Based on the findings of the included studies, only four adult studies provided the prevalence of PH through RHC among patients with elevated ePASP. 22–46% of patients with elevated PASP actually had PH. It should be noted that TTEs are a good screening tool but do not necessarily equate to PH. Furthermore, considering the trim and fill analysis, the prevalence of elevated ePASP in clinically stable SCD patients is sensitive to publication bias, so future systematic review and meta-analysis should consider more comprehensive search to address this issue. Last but not least, our research suffers from language bias and searching of grey literature. Nevertheless, our study possesses several significant strengths. This meta-analysis integrates data from 79 studies selected based on objective criteria, making it the largest investigation to date into the prevalence of elevated ePASP among

SCD patients at steady state worldwide. While SCD is relatively rare, we have taken rigorous measures to avoid the duplication of publications from overlapping study populations.

## Conclusion

This review indicates that the prevalence of elevated ePASP is notably high among certain populations. The public health implications of our findings are substantial, given the potential number of individuals at risk for PH. Further research is essential to elucidate the prevalence and associated factors of elevated ePASP in SCD patients. Additionally, community-based studies are necessary to further characterize the epidemiology and natural history of elevated ePASP in SCD patients. To address this significant health challenge, policymakers and healthcare providers must acknowledge this issue and commit resources toward improving prevention, detection, and management strategies for elevated ePASP.

## Supporting information

**S1 File. The Preferred Reporting Items for Systematic Reviews and Meta-Analyses (PRISMA) 2020 checklist.**
(DOCX)

**S2 File. Search strategies and search results.**
(DOC)

**S3 File. List of excluded studies during initial screening.**
(DOCX)

**S4 File. List of excluded studies at full text review.**
(DOCX)

**S5 File. Risk of bias assessment.**
(DOC)

**S6 File. eTables (eTables 1–6).**
(DOCX)

**S7 File. eFigures (eFigs 1–47).**
(DOC)

## Acknowledgments

Hereby, we sincerely thank all the collaborators who helped us advance this project.

## Author contributions

**Conceptualization:** Mobin Ghazaiean.

**Data curation:** Mobin Ghazaiean, Hadi Darvishi-Khezri, Behnam Najafi, Hossein Karami, Mehrnoush Kosaryan.

**Formal analysis:** Hadi Darvishi-Khezri.

**Investigation:** Mobin Ghazaiean, Hadi Darvishi-Khezri, Hossein Karami, Mehrnoush Kosaryan.

**Methodology:** Mobin Ghazaiean.

**Project administration:** Mobin Ghazaiean.

**Resources:** Mobin Ghazaiean.

**Supervision:** Mobin Ghazaiean.

**Validation:** Mobin Ghazaiean, Hadi Darvishi-Khezri, Hossein Karami, Mehrnoush Kosaryan.

**Visualization:** Mobin Ghazaiean, Hadi Darvishi-Khezri, Behnam Najafi, Hossein Karami, Mehrnoush Kosaryan.

**Writing – original draft:** Mobin Ghazaiean, Hadi Darvishi-Khezri, Behnam Najafi.

**Writing – review & editing:** Mobin Ghazaiean, Hadi Darvishi-Khezri, Behnam Najafi, Hossein Karami, Mehrnoush Kosaryan.

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
