## [Decision Letter · Decision Letter 0]

22 Oct 2024

PONE-D-24-41837Global Prevalence of Elevated Estimated Pulmonary Artery Systolic Pressure in Clinically Stable Pediatrics and Adults with Sickle Cell Disease: A Systematic Review and Meta-analysisPLOS ONE

Dear Dr. Ghazaiean,

Thank you for submitting your manuscript to PLOS ONE. After careful consideration, we feel that it has merit but does not fully meet PLOS ONE’s publication criteria as it currently stands. Therefore, we invite you to submit a revised version of the manuscript that addresses the points raised during the review process. The reviewers found merit in your study but also identified several areas that need more clarification.  For example, the sickle cell disease genotype and definitions of severity that you have used are not consistent with the literature and should be revised accordingly.  Further discussion is also needed to better understand changes before 2014 and after 2015, particularly since hydroxyurea was not recommended until after this time period.

We look forward to receiving your revised manuscript.

Kind regards,

Santosh L. Saraf

Academic Editor

PLOS ONE

Journal Requirements:

1. When submitting your revision, we need you to address these additional requirements. Please ensure that your manuscript meets PLOS ONE's style requirements, including those for file naming. The PLOS ONE style templates can be found at https://journals.plos.org/plosone/s/file?id=wjVg/PLOSOne_formatting_sample_main_body.pdf and https://journals.plos.org/plosone/s/file?id=ba62/PLOSOne_formatting_sample_title_authors_affiliations.pdf 2. Thank you for stating the following in your Competing Interests section: "None" Please complete your Competing Interests on the online submission form to state any Competing Interests. If you have no competing interests, please state "The authors have declared that no competing interests exist", as detailed online in our guide for authors at http://journals.plos.org/plosone/s/submit-now  This information should be included in your cover letter; we will change the online submission form on your behalf. 3. We note that there is identifying data in the Supporting Information file <eTables and S2 Supplementary file>. Due to the inclusion of these potentially identifying data, we have removed this file from your file inventory. Prior to sharing human research participant data, authors should consult with an ethics committee to ensure data are shared in accordance with participant consent and all applicable local laws. Data sharing should never compromise participant privacy. It is therefore not appropriate to publicly share personally identifiable data on human research participants. The following are examples of data that should not be shared: -Name, initials, physical address-Ages more specific than whole numbers-Internet protocol (IP) address-Specific dates (birth dates, death dates, examination dates, etc.)-Contact information such as phone number or email address-Location data-ID numbers that seem specific (long numbers, include initials, titled “Hospital ID”) rather than random (small numbers in numerical order) Data that are not directly identifying may also be inappropriate to share, as in combination they can become identifying. For example, data collected from a small group of participants, vulnerable populations, or private groups should not be shared if they involve indirect identifiers (such as sex, ethnicity, location, etc.) that may risk the identification of study participants. Additional guidance on preparing raw data for publication can be found in our Data Policy (https://journals.plos.org/plosone/s/data-availability#loc-human-research-participant-data-and-other-sensitive-data) and in the following article: http://www.bmj.com/content/340/bmj.c181.long. Please remove or anonymize all personal information (<specific identifying information in file to be removed>), ensure that the data shared are in accordance with participant consent, and re-upload a fully anonymized data set. Please note that spreadsheet columns with personal information must be removed and not hidden as all hidden columns will appear in the published file. 4. As required by our policy on Data Availability, please ensure your manuscript or supplementary information includes the following:  A numbered table of all studies identified in the literature search, including those that were excluded from the analyses.   For every excluded study, the table should list the reason(s) for exclusion.   If any of the included studies are unpublished, include a link (URL) to the primary source or detailed information about how the content can be accessed.  A table of all data extracted from the primary research sources for the systematic review and/or meta-analysis. The table must include the following information for each study:  Name of data extractors and date of data extraction  Confirmation that the study was eligible to be included in the review.   All data extracted from each study for the reported systematic review and/or meta-analysis that would be needed to replicate your analyses.  If data or supporting information were obtained from another source (e.g. correspondence with the author of the original research article), please provide the source of data and dates on which the data/information were obtained by your research group.  If applicable for your analysis, a table showing the completed risk of bias and quality/certainty assessments for each study or outcome.  Please ensure this is provided for each domain or parameter assessed. For example, if you used the Cochrane risk-of-bias tool for randomized trials, provide answers to each of the signalling questions for each study. If you used GRADE to assess certainty of evidence, provide judgements about each of the quality of evidence factor. This should be provided for each outcome.   An explanation of how missing data were handled.   This information can be included in the main text, supplementary information, or relevant data repository. Please note that providing these underlying data is a requirement for publication in this journal, and if these data are not provided your manuscript might be rejected. 5. Please include captions for your Supporting Information files at the end of your manuscript, and update any in-text citations to match accordingly. Please see our Supporting Information guidelines for more information: http://journals.plos.org/plosone/s/supporting-information.

Reviewers' comments:

Reviewer's Responses to Questions

**Comments to the Author**

1. Is the manuscript technically sound, and do the data support the conclusions?

Reviewer #1: Partly

Reviewer #2: Yes

Reviewer #3: Yes

2. Has the statistical analysis been performed appropriately and rigorously? 

Reviewer #1: Yes

Reviewer #2: Yes

Reviewer #3: I Don't Know

3. Have the authors made all data underlying the findings in their manuscript fully available?

Reviewer #1: Yes

Reviewer #2: Yes

Reviewer #3: Yes

4. Is the manuscript presented in an intelligible fashion and written in standard English?

Reviewer #1: Yes

Reviewer #2: Yes

Reviewer #3: Yes

5. Review Comments to the Author

Reviewer #1: Thank you for your efforts to clarify an unclear topic of real significance to the morbidity and mortality of sickle cell disease. Overall, your analysis is well-done, but the groupings do not fully make sense. Specifically sickle cell disease is the overarching diagnosis that includes severe genotypes (HbSS and HbS/beta-zero thalassemia) and these severe genotypes are referred to as "sickle cell anemia." These patients make up the majority of patients with the disease. I have stated this further and with additional comments below.

Abstract

Background: Suggest rephrasing "stable pediatric and adult" as "stable children and adults"

Results/methods: pediatric patients and patients under 18 years old are usually considered to be the same. I don't see this explained in the methods or presented in the results in the text with this same duplicative description

Introduction:

- Both sentences mentioning homozygous vs compound heterozygous sickle cell disease are redundant

- Regarding the statement about the importance of screening and early detection on page 3, the sickle cell guidelines are mixed regarding the utility of screening ECHOs. The American Society of Hematology does not recommend screening ECHO's for asymptomatic patients. Because of this, even though you include "stable patients" it is not possible to know from your analysis if they are asymptomatic. Screening symptomatic patients would yield higher rates of ePASP, and this difference in testing may explain some of the heterogeneity. If the authors are unable to clarify how many patients were symptomatic, please add this as a limitation

Methods

- Why did you include "sickle cell trait" in your search? I would think this should be excluded

- There may be a typo in the description of the study dichotomy by dates "before 2015 and after 2014"?

- The two different groups of "type of disease (SCA and SCD)" vs "genotype severity of SCD" does not make sense to me. Sickle cell anemia typically is used to describe the severe SCD genotypes (HbSS and HbS/beta zero thalassemia). These seem redundant.

- Additionally, patients with SCA (aka with more severe genotypes) are also patients with SCD (which includes all genotypes)

- I would suggest the authors revisit their analysis and either redefine their groups appropriately or remove the attempt to stratify studies based on genotype/disease severity, since patients with severe genotype SCD (aka with SCA) represent the majority of patients with sickle cell disease

Results

- "western" and "non-western" countries does not provide much meaning to readers. Consider stating the countries or continents.

- 3.3 "prevalence of elevated ePASP in adults and pediatrics." Pediatric is an adjective. Consider changing pediatrics to children or adding the word patients to the end

Reviewer #2: Summary:

This manuscript by Ghazaiean et al is a thorough and extensive systematic review and meta-analysis evaluating the prevalence of elevated estimated pulmonary artery systolic pressure (ePASP) in pediatrics and adults with sickle cell disease (SCD) worldwide. The authors included 79 primary studies comprising 6,256 pediatric and 6,582 adult patients with SCD from 22 countries. They found that the prevalence of elevated ePASP was 21.8% in the pediatric population and 30.6% in adults. There were differences in the prevalence of elevated ePASP in subgroup analyses with age, sex, severity of SCD genotype, and year (before 2015 and after 2014). Those with elevated ePASP had significant differences in clinical characteristics and laboratory values compared to those without elevated ePASP. The authors concluded that there is a high prevalence of elevated ePASP in patients with SCD.

Elevated ePASP on a TTE is an important approach to early detection/screening of pulmonary hypertension, a major risk factor for morbidity and mortality in SCD. There was a systematic review on this in 2015 (Caughey et al, BJH), and this updated version 9 years later features the addition of 34 more studies (2,925 more adults & 3,804 more pediatric patients). This is a much needed updated review given there have been various improvements in the management of SCD including in treatment and technology. Given that there is no consensus on optimal screening practices for PH, it is important to first establish the scope of elevated ePASP in the SCD population. Thus, it is critical in the field of SCD to publish this data given the implications for health policy and updated guidelines. The strengths of this paper are its thoroughness, great tables and figures, and interesting subgroup analyses. In the discussion section, there was also a comprehensive review on the pathophysiology of pulmonary hypertension. However, there are some areas where this manuscript could be strengthened, to further enhance and convey the importance of elevated ePASP screening.

Major issues:

• Please cite the previous meta-analysis on ePASP and sickle cell disease (Caughey MC, Poole C, Ataga KI, Hinderliter AL. Estimated pulmonary artery systolic pressure and sickle cell disease: a meta-analysis and systematic review. Br J Haematol. 2015 Aug;170(3):416-24. doi: 10.1111/bjh.13447. Epub 2015 Apr 9. PMID: 25854714.) The authors mention this paper in the introduction and it is a crucial cutoff point in the analysis (before 2015, after 2014), so it should be acknowledged. This paper can also be referenced to highlight in the discussion any new findings from the analysis, and possible explanations. It is great that this work builds on work done previously.

• The definitions for sickle cell anemia (SCA) versus sickle cell disease (SCD) and mild/moderate/severe TRV, should be clearly defined as it is not entirely apparent who is included in these groups, which makes the paper confusing to read. Does SCA just include HbSS or did it also include HbSbeta-0-thalassemia? For severity of ePASP, what were the cutoffs? These definitions are important as the authors later make recommendations on screening practices.

• The authors have many interesting findings such as the decrease in prevalence of ePASP in pediatrics after 2014 compared to before 2015, lab correlations, sex differences, and non-Western countries versus Western countries – the manuscript could be enhanced with further discussion of at least a few of these findings if not all of them. TTE may be more accessible in non-Western countries, which may be an important factor as to why this is an important screening tool.

Minor issues:

• In the introduction paragraph, the following sentence is ambiguous to me as it makes right heart catheterizations seem like the initial and sole method of diagnosing pulmonary hypertension for some cases, but it is typically done after a screening TTE: “This underscores the importance of screening and early detection of PASP elevations through non-invasive methods, such as trans-thoracic echocardiography, rather than relying solely on invasive right heart catheterization (RHC) for diagnosis.”

• In the introduction, when mentioning the compound heterozygous state, you should clarify that HbSC is an example of a compound heterozygous state, as the way it is written know suggests it is the only heterozygous state.

• In the introduction, would clarify that a RHC is required for diagnosis of PH. TRV > 2.5 is a great screening tool but is not always indicative of pulmonary hypertension (sensitivity increases with higher TRV), however, given that elevated TRV is associated with increased mortality, it is still an important clinical feature to assess, particularly since it is noninvasive.

• It feels as though the author of sections 3.3.1-3.3.2 was different from the author of 3.3.3- 3.3.7. The use of the phrase, “meticulously assessed” in the latter sections do not appear consistent with the writing style of previous sections. Would recommend being more consistent and possibly removing a few of the “meticulously” phrases as it is repetitive.

• In section 3.4, please highlight some important takeaways from table 3.

• In the discussion, you mention, “Thankfully, there has been a slightly meaningful decrease in the prevalence of elevated ePASP in children since 2015, as compared to the previous rate.” Why is this 2015 date important? The meta-analysis had been published at that time, but those results would not change clinical practice. What do you think could explain this finding?

• In the discussion, you write “The prevalence of elevated ePASP is notably higher in pediatric males compared to females. However, this trend reverses in adults, where females exhibit a greater prevalence than males.” Please discuss why this is important or what could explain this.

• In the discussion, you write, “Clinical and laboratory analyses of pediatric and adult patients with SCD revealed significant mean differences in various characteristics, including age, oxygen saturation, Hb, HbF, WBC count, and reticulocyte levels between those with elevated ePASP and those without.” Please discuss why this is important.

• On the last sentence of page 26: “Our study proposes a noninvasive screening protocol for PH in children with SCD, which may facilitate the identification of a specific subgroup of patients who would benefit from further evaluation via catheterization. Implementing a strategy to recognize children at risk for hemolysis-associated vasculopathy through a straightforward biomarker, such as elevated TRV, followed by preventive interventions like hydroxycarbamide, transfusions, or other methods to mitigate hemolysis, could significantly enhance survival outcomes in SCD.” – what is your screening protocol? Screen every pediatric patient with TTE? What age? How often? Is there a TRV cutoff? Should we follow the ATS guidelines or ASH guidelines? It would be important to acknowledge there is no consensus. TRV is not a straightforward biomarker, as you mention later in the limitation section – TTEs are highly tech and reader dependent. It would also be important to mention the next steps if the TRV is elevated, including obtaining a pro-NT-BNP, assessing 6 minute walk distance, and a RHC. It would be interesting to discuss any differences in the approach of non-Western versus Western countries given the amount of resources in each setting. It was thoughtful to include potential treatment strategies.

• In the limitations section, I would not recommend listing each limitation with “firstly,” “secondly,” “thirdly,” … “twelfthly” as this makes the paragraph choppy and less interesting to read.

• Section 2.7 Risk of bias assessment – please write in past tense, as you did for the other paragraphs in this section.

• Table 3 is great. Why is “No of studies/total listed” twice?

• I am surprised that there were no differences observed between patients from Western and non-Western countries? Why do you think that is?

Reviewer #3: Abstract

1. In the results section, can the authors clarify the definition of SCA and severe genotype? SCA and severe genotype both typically refer to HbSS and HbSbeta zero thalassemia but both have different prevalence rates and confidence intervals. The statement below is from the paper…

“When stratified by disease type, the prevalence rates for patients with sickle cell anemia (SCA) were 16.55% (95% CI: 12.49 to 20.61) in pediatric patients and 36.06% (95% CI: 31.53 to 40.6) in adults. The prevalence of elevated ePASP among studies with severe SCD genotype was found to be 19.45% (95% CI: 14.95 to 23.95) in pediatric patients and 29.55% (95% CI: 24.21 to 34.89) in adults.”

Introduction

1. Can the authors elaborate more on the importance of elevated PASP in the SCD population in the introduction? For example, multiple studies exist linking elevated TRV with increased morbidity in children and mortality in adults. This was more clearly explained in the discussion but would have been helpful in the introduction to carry the reader along.

2. Sickle cell anemia (SCA) conventionally refers to HbSS and HbSbeta zero thalassemia. Also referred to as “severe genotype.” However, the authors report different values for SCA and “severe genotype.” Please clarify definition.

Results

1. Since the study involved such robust statistical analysis, was a biostatistician involved in this study? I do not see a biostatistician listed among the authors.

2. Were there any echo complications seen in those patients with elevated PASP like diastolic dysfunction? Or were studies with these complications completely excluded during the review process?

- Were other parameters such as normal ejection fraction, left ventricular wall thickness, etc evaluated?

3. Did any cohort have right heart catheterization data? It will be helpful to include this data to demonstrate a correlation between elevated PASP confirmed by right heart catheterization.

4. I found it interesting that there was no statistically significant difference in the prevalence rates of elevated PASP across Western and non-Western countries. Can the authors provide a possible reason for this?

5. Why were the date ranges selected for analysis? “Studies published prior to 2014” and “prior to 2015” were specifically selected as the cut-off year in the selection.

Discussion

1. Can the authors put forward a hypothesis why there has been a “slightly meaningful decrease” in the prevalence of elevated PASP in children since 2015? It is interesting because hydroxyurea did not become very widely adopted until FDA approved hydroxyurea for use in children in 2017. Other factors may be in play.

2. The paper ostensibly sought to estimate the global prevalence of elevated estimated pulmonary artery systolic pressure in children and adults with SCD, but the discussion devotes large sections to tricuspid regurgitation velocity (TRV). Is the argument that PASP may be superior to TRV in estimating pulmonary hypertension or are they equally acceptable alternatives? Does one measurement parameter have any advantage over the other specifically for estimating PH in SCD?

4. Instead of using Firstly to twelfthly to highlight limitations, perhaps the authors can combine some limitations and use other introductory words such as “Furthermore” to introduce the other limitations.

6. PLOS authors have the option to publish the peer review history of their article (what does this mean? ). If published, this will include your full peer review and any attached files.

**Do you want your identity to be public for this peer review?** For information about this choice, including consent withdrawal, please see our Privacy Policy .

Reviewer #1: No

Reviewer #2: No

Reviewer #3: No

---

## [Author Response · Author response to Decision Letter 0]

11 Dec 2024

Date: 2024-11-21

Subject: Response to Reviewer's Comments

Title: PLOS ONE

ID: PONE-D-24-41837

Dear Editor

I sincerely appreciate the editor and reviewers. We believe that the manuscript has been improved. We hope that the revised manuscript has achieved the journal's standards. We submitted the revised manuscript with the reply to the editor and reviewers' comments as point-to-point.

Editor Comments to Author:

The reviewers found merit in your study but also identified several areas that need more clarification. For example, the sickle cell disease genotype and definitions of severity that you have used are not consistent with the literature and should be revised accordingly. Further discussion is also needed to better understand changes before 2014 and after 2015, particularly since hydroxyurea was not recommended until after this time period.

Response: Thank you so much for your comment and attention. I would like to address this misunderatanding that caused for the editor and all the reviewers. The decision for considering the subgroups in the current study was based on the definition of each subgroup. The subgroups including “type of patients” and “severe genotype (HbSS and HbS/β0)” was made in order to assess these specific SCD populations accurately. Some use the term ‘‘sickle cell anemia’’ to include both the HbSS and HbS/β0 genotypes, however, based on the definition in the literature, HbS/β0 thalassemia can be distinguished from sickle cell anemia by analyzing the morphology of red blood cells and the results of hemoglobin electrophoresis [1-3]. So, we considered the subgroup based on the definition in the literature. Also, we considered the subgroup severe genotypes (HbSS and HbS/β0) in order to specifically present the prevalence of elevated ePASP in these genotypes and we did not include the studies that did not define the genotypes through hemoglobin electrophoresis.

In order to resolve this misunderstanding that can also cause for the readers, we decided to delete the subgroup “Type of disease” from the entire of the study [we corrected the order of the eFigures in the manuscript] and supplementary file (eTables and eFigures) and we specifically focused on the studies clearly defined the most severe forms of SCD, HbSS and HbS/β0 through genotyping.

The definitions of all subgroups were explained in the Method section (Lines 162-176).

We discussed the changes before 2014 and after 2015 in the Discussion section (Lines 474-488 and 469-473).

We revised the severity of elevated ePASP in adults (Table 1).

Response: Thank you so much for your comment. We modified our manuscript based on these templates.

2. Thank you for stating the following in your Competing Interests section: "None"

Please complete your Competing Interests on the online submission form to state any Competing Interests. If you have no competing interests, please state "The authors have declared that no competing interests exist", as detailed online in our guide for authors at http://journals.plos.org/plosone/s/submit-now

Response: Thank you so much for your comment. We completed our Competing Interests on the online submission form. We stated th competing interest both in the manuscript (Lines 600-601) and cover letter.

3. We note that there is identifying data in the Supporting Information file <eTables and S2 Supplementary file>. Due to the inclusion of these potentially identifying data, we have removed this file from your file inventory. Prior to sharing human research participant data, authors should consult with an ethics committee to ensure data are shared in accordance with participant consent and all applicable local laws.

-Location data

Response: Thank you so much for your comment. As you stated, we removed the columns with personal information from the supplenetary files (including eTables and eFigures files) and modified our files based on the mentioned changes. Notably, the current S4 file (including a list of excluded studies at full text review with brief report) did not contain any identifying data.

4. As required by our policy on Data Availability, please ensure your manuscript or supplementary information includes the following:

Response: Thank you so much for your comment. We provided numbered table of all studies identified in the literature search. Also, we provided a table for the excluded studies with the reason for exclusion in Supporting information file, “S3 File. List of excluded studies”. All the studies were published. We provided tables of all data extracted from the primary research sources in Supporting information file, “S6 File. eTables (eTables 1 - 6)”. We mentioned the data extractors, date of data extraction, the eligibility of the included studies in the manuscript. All data or supporting information were obtained from the included studies. We provided the completed risk of bias and quality/certainty assessments for each study in Supporting information file, “S4 File. Risk of bias assessment”. We tried to handle the missing data of the outcome of interest through risk of bias assessment which we devoted two scores of the modified version of the checklist regarding the missing data including item 5 and item 8 of the checklist (Lines 155-156).

Response: Thank you so much for your comment. We provided captions for our Supporting Information files at the end of the manuscript, and updateed the in-text citations (Lines 576-583).

Reviewers'comments:

Reviewer's Responses to Questions

Comments to the Author

1. Is the manuscript technically sound, and do the data support the conclusions?

Reviewer #1: Partly

Reviewer #2: Yes

Reviewer #3: Yes

Response: Thank you so much for your comment and attention. Based on the editor and reviewers’ comments, we did our best to address those comments and we believe that the manuscript has been improved. We have highlighted all the changes in yellow. We hope that the revised manuscript has achieved the journal's standards.

2. Has the statistical analysis been performed appropriately and rigorously?

Reviewer #1: Yes

Reviewer #2: Yes

Reviewer #3: I Don't Know

Response: Thank you so much for your attention.

3. Have the authors made all data underlying the findings in their manuscript fully available?

Reviewer #1: Yes

Reviewer #2: Yes

Reviewer #3: Yes

Response: Thank you so much for your comment.

4. Is the manuscript presented in an intelligible fashion and written in standard English?

Reviewer #1: Yes

Reviewer #2: Yes

Reviewer #3: Yes

Response: Thank you so much for your consideration.

Reviewer Comments to Author:

Reviewer 1:

Comment: Thank you for your efforts to clarify an unclear topic of real significance to the morbidity and mortality of sickle cell disease. Overall, your analysis is well-done, but the groupings do not fully make sense. Specifically sickle cell disease is the overarching diagnosis that includes severe genotypes (HbSS and HbS/beta-zero thalassemia) and these severe genotypes are referred to as "sickle cell anemia." These patients make up the majority of patients with the disease. I have stated this further and with additional comments below.

Response: Thank you so much for your comment and attention. I would like to address this misunderatanding that caused for the editor and all the reviewers. The decision for considering the subgroups in the current study was based on the definition of each subgroup. The subgroups including “type of patients” and “severe genotype (HbSS and HbS/β0)” was made in order to assess these specific SCD populations accurately. Some use the term ‘‘sickle cell anemia’’ to include both the HbSS and HbS/β0 genotypes, however, based on the definition in the literature, HbS/β0 thalassemia can be distinguished from sickle cell anemia by analyzing the morphology of red blood cells and the results of hemoglobin electrophoresis [1-3]. So, we considered the subgroup based on the definition in the literature. Also, we considered the subgroup severe genotypes (HbSS and HbS/β0) in order to specifically present the prevalence of elevated ePASP in these genotypes and we did not include the studies that did not define the genotypes through hemoglobin electrophoresis.

In order to resolve this misunderstanding that can also cause for the readers, we decided to delete the subgroup “Type of disease” from the entire of the study [we corrected the order of the eFigures in the manuscript] and supplementary file (eTables and eFigures) and we specifically focused on the studies clearly defined the most severe forms of SCD, HbSS and HbS/β0 through genotyping.

The definitions of all subgroups were explained in the Method section (162-176).

Comment: Abstract

Background: Suggest rephrasing "stable pediatric and adult" as "stable children and adults"

Results/methods: pediatric patients and patients under 18 years old are usually considered to be the same. I don't see this explained in the methods or presented in the results in the text with this same duplicative description

Response: Thank you so much for your attention. "stable pediatric and adult" substituted by "stable children and adults" in the Background subsection (Lines 26-27) and throughout the manuscript.

We also mentioned the definition of children and adults in the Abstract (Line 34) and Method sections (Lines 163-164).

Comment: Introduction

- Both sentences mentioning homozygous vs compound heterozygous sickle cell disease are redundant

Response: Thank you so much for your attention. We deleted these sentences.

Comment:

- Regarding the statement about the importance of screening and early detection on page 3, the sickle cell guidelines are mixed regarding the utility of screening ECHOs. The American Society of Hematology does not recommend screening ECHO's for asymptomatic patients. Because of this, even though you include "stable patients" it is not possible to know from your analysis if they are asymptomatic. Screening symptomatic patients would yield higher rates of ePASP, and this difference in testing may explain some of the heterogeneity. If the authors are unable to clarify how many patients were symptomatic, please add thi

---

## [Decision Letter · Decision Letter 1]

8 Jan 2025

PONE-D-24-41837R1Global Prevalence of Elevated Estimated Pulmonary Artery Systolic Pressure in Clinically Stable Children and Adults with Sickle Cell Disease: A Systematic Review and Meta-analysisPLOS ONE

Dear Dr. Ghazaiean,

Thank you for submitting your manuscript to PLOS ONE. After careful consideration, we feel that it has merit but does not fully meet PLOS ONE’s publication criteria as it currently stands. Therefore, we invite you to submit a revised version of the manuscript that addresses the points raised during the review process.

There are still some minor issues that the reviewer has indicated that should be addressed to improve the clarity of the manuscript.  

We look forward to receiving your revised manuscript.

Kind regards,

Santosh L. Saraf

Academic Editor

PLOS ONE

Journal Requirements:

Reviewers' comments:

Reviewer's Responses to Questions

**Comments to the Author**

1. If the authors have adequately addressed your comments raised in a previous round of review and you feel that this manuscript is now acceptable for publication, you may indicate that here to bypass the “Comments to the Author” section, enter your conflict of interest statement in the “Confidential to Editor” section, and submit your "Accept" recommendation.

Reviewer #2: (No Response)

Reviewer #3: All comments have been addressed

2. Is the manuscript technically sound, and do the data support the conclusions?

Reviewer #2: Yes

Reviewer #3: Yes

3. Has the statistical analysis been performed appropriately and rigorously? 

Reviewer #2: Yes

Reviewer #3: Yes

4. Have the authors made all data underlying the findings in their manuscript fully available?

Reviewer #2: Yes

Reviewer #3: Yes

5. Is the manuscript presented in an intelligible fashion and written in standard English?

Reviewer #2: Yes

Reviewer #3: Yes

6. Review Comments to the Author

Reviewer #2: The revised manuscript is greatly improved although there are a few small points that should be addressed prior to being published. Here are my comments below.

1. In the abstract, lines 39-41 and the second conclusion (“There has been a slightly meaningful reduction in the prevalence of elevated ePASP among children since 2014) is an overstatement, so I would recommend removing these, particularly since it was not explained in the discussion.

2. In the abstract, lines 37-38, “Age specific analysis… in adults” does not add to the results section and can be removed.

3. In the introduction, lines 64-72, the authors wrote: “The TRV cutoff of >=2.5m/s seems inappropriate… (It is important to note that NT-pro-BNP levels can be deceptive in a condition of renal insufficiency)” should not be in the introduction. Discussing the ambiguity of TRV measurements and cutoffs in the introduction weakens the premise of this meta-analysis (that people use TRV measurements to estimate ePASP), and should instead be in the limitations section.

4. In the introduction and the rest of the manuscript, please be consistent with “TRV” and “TRJV” – are these the same? Only TRJV was defined in line 64 as tricuspid regurgitant jet velocity. Also please define the abbreviation ePASP in the introduction as well – PASP was defined in line 62, but ePASP appears in 87 and the reader may not know what the “e” stands for.

5. In the definition section, line 167, what is SPAP? Please define.

6. For tables 1 and 2, they would be strengthened by including the number of patients used to calculate the prevalence rate. You mention using thousands of patients in the meta-analysis, so it would strengthen the validity of the table to show these numbers here as well.

7. For the results section, I am a little perplexed by the sex-specific analysis. Are the authors saying that among male children, the rate of elevated ePASP is 60.35% (because that is quite alarming) or are the authors saying that among children with elevated ePASP, the percentage of males is 60.35%? Similarly, for the adults, is the prevalence of elevated ePASP among male adults with SCD 45.6%? - because that is what is implied by the table. The discussion on sex (lines 452-461) may have to be adjusted if the prevalence rates change upon further re-analysis.

8. Please be consistent in what is statistically significant. If your cutoff is p<=0.05, please do not write that p=0.06 is marginally significant, as this can be misleading. Instead, for the analyses where p=0.06 (variation in heterogeneity indices, children’s ePASP prevalence before and after 2015), you can say that numerically there was a trend towards some specific direction, but it was not statistically significant.

9. In table 3, why are the P-values of the mean difference and standardized mean difference columns separated by the columns “heterogeneity/ I-square and p-value”? If the model is the same for all of the studies, this column can be removed.

10. In the discussion, lines 422-424, the authors note that 22-46% of patients with elevated PASP actually had PH. This should be highlighted as a limitation of using ePASP from TTEs – that TTEs are a good screening tool but do not necessarily equate to pulmonary hypertension. However, there are studies, as the author mentioned that elevated TRVs do portend worse outcomes, so this can be highlighted as well.

11. In the discussion, lines 474-488: this paragraph is confusing and hard to follow. The first sentence states that the prevalence of elevated ePASP in children decreased after 2015, so I was expecting a paragraph about why the authors thought this finding could be valid and true. However, it is unclear how HbF and other factors that affect disease severity in SCD relate to a lower prevalence of ePASP in children, specifically.

12. The discussion should also highlight the interesting findings from the subgroup analyses and how these factors could confound the TTE findings (severe anemia, low O2, etc) or how these could potentially help figure out who to screen (elevated WBC, low Hgb, low HbF, etc). These markers are all indicators of severe disease, and given that the authors consolidated multiple studies, the fact that these all are associated with elevated ePASP is validating.

Reviewer #3: (No Response)

7. PLOS authors have the option to publish the peer review history of their article (what does this mean? ). If published, this will include your full peer review and any attached files.

**Do you want your identity to be public for this peer review?** For information about this choice, including consent withdrawal, please see our Privacy Policy .

Reviewer #2: No

Reviewer #3: **Yes: ** Chibuzo Ilonze

---

## [Author Response · Author response to Decision Letter 1]

16 Jan 2025

Date: 2025-01-16

Subject: Response to Reviewer's Comments

Title: PLOS ONE

ID: PONE-D-24-41837R1

Dear Editor

I sincerely appreciate the editor and reviewers. We believe that the manuscript has been improved. We hope that the revised manuscript has achieved the journal's standards. We submitted the revised manuscript with the reply to the editor and reviewers' comments as point-to-point.

Journal Requirements:

Response: Thank you so much for your comment and attention. We reviewed the reference list that was complete and correct. No cited papers have been retracted. According to the reviewer’s comment regarding the improvement in some parts of the Discussion section, some references added to the manuscript which have been highlighted in yellow in the “References section”. (Lines 767 to 805 and Lines 821 to 832 and Lines 874 to 896 and Lines 899 to 900 and Lines 902 to 906 and Lines 909 to 917)

Reviewers' comments:

Reviewer's Responses to Questions

Comments to the Author

1. If the authors have adequately addressed your comments raised in a previous round of review and you feel that this manuscript is now acceptable for publication, you may indicate that here to bypass the “Comments to the Author” section, enter your conflict of interest statement in the “Confidential to Editor” section, and submit your "Accept" recommendation.

Reviewer #2: (No Response)

Reviewer #3: All comments have been addressed

Response: Thank you so much for your comment. We hope that the revised manuscript has achieved the journal's standards.

2. Is the manuscript technically sound, and do the data support the conclusions?

Reviewer #2: Yes

Reviewer #3: Yes

Response: Thank you so much for your comment.

3. Has the statistical analysis been performed appropriately and rigorously?

Reviewer #2: Yes

Reviewer #3: Yes

Response: Thank you so much for your comment.

4. Have the authors made all data underlying the findings in their manuscript fully available?

Reviewer #2: Yes

Reviewer #3: Yes

Response: Thank you so much for your comment.

5. Is the manuscript presented in an intelligible fashion and written in standard English?

Reviewer #2: Yes

Reviewer #3: Yes

Response: Thank you so much for your comment.

6. Review Comments to the Author

Reviewer #2: The revised manuscript is greatly improved although there are a few small points that should be addressed prior to being published. Here are my comments below.

1. In the abstract, lines 39-41 and the second conclusion (“There has been a slightly meaningful reduction in the prevalence of elevated ePASP among children since 2014) is an overstatement, so I would recommend removing these, particularly since it was not explained in the discussion.

Response: Thank you so much for your comment and attention. We removed these sentences.

2. In the abstract, lines 37-38, “Age specific analysis… in adults” does not add to the results section and can be removed.

Response: Thank you so much for your attention. We agree and we removed this sentence from the result subsection of the Abstract section.

3. In the introduction, lines 64-72, the authors wrote: “The TRV cutoff of >=2.5m/s seems inappropriate… (It is important to note that NT-pro-BNP levels can be deceptive in a condition of renal insufficiency)” should not be in the introduction. Discussing the ambiguity of TRV measurements and cutoffs in the introduction weakens the premise of this meta-analysis (that people use TRV measurements to estimate ePASP), and should instead be in the limitations section.

Response: Thank you so much for your comment and consideration. We removed these sentences from the introduction section and added them to the Limitation section. (Lines 603 to 607)

4. In the introduction and the rest of the manuscript, please be consistent with “TRV” and “TRJV” – are these the same? Only TRJV was defined in line 64 as tricuspid regurgitant jet velocity. Also please define the abbreviation ePASP in the introduction as well – PASP was defined in line 62, but ePASP appears in 87 and the reader may not know what the “e” stands for.

Response: Thank you so much for your consideration. We resolved this misunderstanding through the whole manuscript which is highlighted in yellow. We also defined the abbreviation of ePASP in the introduction. We added the abbreviation to the Abbreviation section. (Line 75 and Line 632)

5. In the definition section, line 167, what is SPAP? Please define.

Response: Thank you so much for your comment and attention. We defined SPAP (systolic pulmonary artery pressure) in the definition section. We also added the abbreviation to the Abbreviation section. (Line 155 and Line 633)

6. For tables 1 and 2, they would be strengthened by including the number of patients used to calculate the prevalence rate. You mention using thousands of patients in the meta-analysis, so it would strengthen the validity of the table to show these numbers here as well.

Response: Thank you so much for your attention. We added new columns to the both Tables 1 and 2 called “No of patients/total”. We also added this footnote “This column represents the number of patients used to calculate the prevalence rate. Detailed information regarding the calculation of the prevalence values of each study is mentioned in S7 File. eFigures” to the both Tables.” (Line 228, Lines 233 to 234, Line 245, and Lines 250 to 251)

7. For the results section, I am a little perplexed by the sex-specific analysis. Are the authors saying that among male children, the rate of elevated ePASP is 60.35% (because that is quite alarming) or are the authors saying that among children with elevated ePASP, the percentage of males is 60.35%? Similarly, for the adults, is the prevalence of elevated ePASP among male adults with SCD 45.6%? - because that is what is implied by the table. The discussion on sex (lines 452-461) may have to be adjusted if the prevalence rates change upon further re-analysis.

Response: Thank you so much for your attention. We used to mention on the heading related to this finding “Prevalence of elevated ePASP based on the age and sex of the patients with elevated ePASP in children and adults” in the Result section (Lines 255 to 256), so we provided the prevalence values related to the sex among SCD patients with elevated ePASP not among the SCD patients. We modified the abstract section. We also added this footnote “The prevalence values based on the sex were calculated among SCD patients with elevated ePASP.” to the both Tables 1 and 2 and also modified the sentence in the Discussion section in order to avoid any misunderstanding. (Lines 37 to 38, Line 232, Line 249, Line 264, Line 270, Line 498, and Line 505)

8. Please be consistent in what is statistically significant. If your cutoff is p<=0.05, please do not write that p=0.06 is marginally significant, as this can be misleading. Instead, for the analyses where p=0.06 (variation in heterogeneity indices, children’s ePASP prevalence before and after 2015), you can say that numerically there was a trend towards some specific direction, but it was not statistically significant.

Response: Thank you so much for your attention. We edited the manuscript based on this comment. (Lines 264 to 265, Line 268, and Lines 345 to 346)

9. In table 3, why are the P-values of the mean difference and standardized mean difference columns separated by the columns “heterogeneity/ I-square and p-value”? If the model is the same for all of the studies, this column can be removed.

Response: Thank you so much for your consideration and attention. The p-value mentioned in the tables 3 and 4 under the column of “Heterogeneity” relates to the “Test of heterogeneity”. To determine whether significant heterogeneity exists, readers should look for the P value of the test of heterogeneity. A high P value is good news because it suggests that the heterogeneity is insignificant and it can guide readers while interpreting the results. We also added footnotes to the Tables 3 and 4. (Lines 384 to 385 and Lines 400 to 401)

10. In the discussion, lines 422-424, the authors note that 22-46% of patients with elevated PASP actually had PH. This should be highlighted as a limitation of using ePASP from TTEs – that TTEs are a good screening tool but do not necessarily equate to pulmonary hypertension. However, there are studies, as the author mentioned that elevated TRVs do portend worse outcomes, so this can be highlighted as well.

Response: Thank you so much for your attention. We added this sentence to the limitation section (Lines 612 to 614). We also discussed the studies that elevated TRVs do portend worse outcomes. (Lines 459 to 473)

11. In the discussion, lines 474-488: this paragraph is confusing and hard to follow. The first sentence states that the prevalence of elevated ePASP in children decreased after 2015, so I was expecting a paragraph about why the authors thought this finding could be valid and true. However, it is unclear how HbF and other factors that affect disease severity in SCD relate to a lower prevalence of ePASP in children, specifically.

Response: Thank you so much for your comment. We modified the mentioned paragraph and we discussed the potential factors as well. (Lines 523 to 568)

12. The discussion should also highlight the interesting findings from the subgroup analyses and how these factors could confound the TTE findings (severe anemia, low O2, etc) or how these could potentially help figure out who to screen (elevated WBC, low Hgb, low HbF, etc). These markers are all indicators of severe disease, and given that the authors consolidated multiple studies, the fact that these all are associated with elevated ePASP is validating.

Response: Thank you so much for your attention. According to your comment, certain researchers have indicated that TRV is probably elevated in patients with severe anemia who have increased cardiac output and high stroke volume [1, 2]. However, the patients of the included studies were in steady state and not to be severely anemic.

We discussed the interesting findings from the subgroup analyses and provided a summary of findings that how these factors could confound the TTE findings. (Lines 414 to 451)

Reviewer #3: (No Response)

Response: Thank you so much for your time and attention.

7. PLOS authors have the option to publish the peer review history of their article (what does this mean?). If published, this will include your full peer review and any attached files.

Response: Thank you so much. According to your comment, we agree to publish the peer review history.

Do you want your identity to be public for this peer review? For information about this choice, including consent withdrawal, please see our Privacy Policy.

Reviewer #2: No

Reviewer #3: Yes: Chibuzo Ilonze

Response: Thank you so much. I have a lot of respect for the time and attention of the editor and the reviewers have given to this manuscript and I wish all of them the best.

Sincerely,

Corresponding author: dr.mobin.gh@gmail.com

Address: Mazandaran University of Medical Science, Bou Ali Sina Hospital, Pasdaran Boulevard, Sari, Mazandaran Province, Iran.

PO BOX: 481675793

Tel & Fax: +98(0)11-33347837, +98(0)11-33349672

Scopus: 57726509000

Researcher ID: JUV-2911-2023

References

1. Bunn, H.F., et al., Pulmonary hypertension and nitric oxide depletion in sickle cell disease. Blood, The Journal of the American Society of Hematology, 2010. 116(5): p. 687-692.

2. Caughey, M.C., et al., Hemodynamic characteristics and predictors of pulmonary hypertension in patients with sickle cell disease. The American journal of cardiology, 2012. 109(9): p. 1353-1357.

---

## [Editor Report · Decision Letter 2]

22 Jan 2025

Global Prevalence of Elevated Estimated Pulmonary Artery Systolic Pressure in Clinically Stable Children and Adults with Sickle Cell Disease: A Systematic Review and Meta-analysis

PONE-D-24-41837R2

Dear Dr. Ghazaiean,

We’re pleased to inform you that your manuscript has been judged scientifically suitable for publication and will be formally accepted for publication once it meets all outstanding technical requirements.

Kind regards,

Santosh L. Saraf

Academic Editor

PLOS ONE
---

## [Editor Report · Acceptance letter]

PONE-D-24-41837R2

PLOS ONE

Dear Dr. Ghazaiean,

I'm pleased to inform you that your manuscript has been deemed suitable for publication in PLOS ONE. Congratulations! Your manuscript is now being handed over to our production team.

Kind regards,

on behalf of

Dr. Santosh L. Saraf

Academic Editor

PLOS ONE